Analysis

# Cell populations in human breast cancers are molecularly and biologically distinct with age

Adrienne Parsons[1,2,3,4], Esther Sauras Colón [1,5], Meghana Manjunath[1,2], Hanyun Zhang [6,7], Julia Chen[6,7], Milos Spasic[1,2], Beyza Koca [2,3,4], Busem Binboga Kurt[2,8,9], Rachel A. Freedman[2,3,4,10], Elizabeth A. Mittendorf [3,4,9,10], Alexander Swarbrick [6,7], Peter van Galen [1,2,11,12,13,14] ✉ & Sandra S. McAllister [1,2,10,11,12,14] ✉

Aging is associated with increased breast cancer risk, and the oldest and youngest patients have worse outcomes, irrespective of subtype. It is unknown how age affects cells in the breast tumor microenvironment or how they contribute to age-related pathology. Here we discover age-associated differences in cell states in human estrogen receptor-positive and triple-negative breast cancers using analyses of existing bulk and single-cell transcriptomic data. We generate and apply an Age-Specific Program ENrichment (ASPEN) analysis pipeline, revealing age-related changes, including increased tumor cell epithelial–mesenchymal transition and cancer-associated fibroblast inflammatory responses in triple-negative breast cancer. Estrogen receptor-positive breast cancer displays increased *ESR1* expression and reduced vascular and immune cell metabolism with age. Cell interactome analysis reveals candidate signaling pathways that drive age-related cell states. Spatial analyses across independent clinical cohorts support the computational findings. This work identifies potential targets for age-adapted therapeutic interventions for breast cancer.

Breast cancer is the second most commonly diagnosed cancer worldwide[1,2]. Relative to patients aged 55–64 years, both younger (<45) and older (>65) patients with early-stage disease have worse breast cancer-related outcomes, regardless of subtype; older patients fare the worst[3,4]. The reasons for high breast cancer mortality for the youngest and oldest patients are unknown. Confounding our understanding is the fact that older patients with breast cancer are underrepresented in clinical trials, despite comprising most breast cancer cases[5,6]. Patients under the age of 40 are also underrepresented in trials because they represent only ~7% of all breast cancer cases[7]. These deficits imply that real-world outcomes may not align with trial results.

Current breast cancer treatments are often modified for different age populations because of tolerability, comorbidities and variable toxicity[8–11], and age at diagnosis differentially affects prognosis depending on molecular subtype[12–14]. Younger patients are at higher risk for aggressive subtypes of breast cancer, like triple-negative breast cancer (TNBC), and young age is an independent risk factor for TNBC recurrence and death[7,15,16]. Breast cancer incidence increases with age; as hormone receptor-positive (HR⁺) disease increases the most dramatically, it represents the most prevalent subtype among older patients[17]. Observations like these suggest that age-related factors underlie breast cancer initiation and progression.

Prior studies, including our own, show that the breast cancer molecular landscape differs with age[11,18]. A subtype-specific understanding of the age-associated molecular programs defining breast cancer at cell-type resolution could provide much needed insights into this

---

emerging area. In this study, we develop a framework for understanding cell-specific, age-associated changes in gene expression, protein levels and intercellular interactions within the tumor microenvironment (TME) in TNBC and estrogen receptor positive (ER)$^+$ breast cancers. We find that age is a strong driver of microenvironment heterogeneity: tumor-associated epithelial, immune and stromal cell types are biologically distinct with age and subtype. Our results define age-associated, subtype-specific molecular and functional programs, suggesting opportunities to develop age-appropriate therapeutic strategies.

## Results

### Age-related gene expression in TNBC and ER$^+$ breast cancer

We first analyzed gene expression in tumors from patients with stage I–III TNBC or ER$^+$/human epidermal growth factor receptor (HER2)$^-$ breast cancer using the Molecular Taxonomy of Breast Cancer International Consortium (METABRIC) and with stage I–III basal or luminal A breast cancer in The Cancer Genome Atlas (TCGA) BRCA bulk gene expression databases[19,20]. We defined age stratifications of less than 45 years ('younger') and more than 65 years ('older') at diagnosis to align with established clinical risk[3,4]. METABRIC samples were identified according to histopathological subtype (TNBC and ER$^+$); TCGA samples included basal and luminal A molecular subtypes to build the age cohorts (Source Data for Fig. 1).

Of the differentially expressed genes (DEGs) in younger versus older patient-derived TNBC/basal tumors (Source Data for Fig. 1), 15 were common across both datasets (Fig. 1a and Extended Data Fig. 1a). For example, genes enriched in tumors from older patients included *EPYC* (encoding epiphycan, which promotes epithelial–mesenchymal transition (EMT) in ovarian cancer[21]) and *SERPINA1* (encoding a serine protease inhibitor associated with poor overall survival in patients with breast cancer[22]).

Among the ER$^+$/luminal A breast tumor DEGs that were common to both datasets, 83 genes were enriched in younger and 32 in older patient tumors (Fig. 1b, Extended Data Fig. 1b and Source Data for Fig. 1). Consistent with a prior report[11], *ESR1* (encoding estrogen receptor 1) was highly enriched in the older cohorts (Fig. 1b), possibly because of an uncontrolled feedback loop driven by low postmenopausal estrogen.

Gene set enrichment analysis (GSEA) revealed that several of the significantly enriched gene sets in tumors from older patients with basal disease/TNBC involved immune processes, including antigen processing and presentation (particularly via major histocompatibility complex (MHC) class II), inflammation response, and interferon-γ (IFNγ) signaling (Fig. 1c, Extended Data Fig. 1c and Source Data for Fig. 1). In the younger TNBC/basal cohorts, gene sets involved in cell cycle and oncogenic signaling were significantly enriched (Fig. 1c).

In ER$^+$/luminal A breast cancer, all DEG sets were enriched in tumors from younger patients and related to breast biology, breast cancer molecular subtype and mitogenic stimuli (Fig. 1d, Extended Data Fig. 1d and Source Data for Fig. 1). These cancers displayed no age-stratified immune responses, except for tumor necrosis factor (TNF) signaling enrichment in the younger cohort (Fig. 1d and Source Data for Fig. 1).

### Development of the Age-Specific Program ENrichment analysis method

Leveraging the resolution and deeper insights afforded by single-cell transcriptomics, we developed an analysis pipeline, termed Age-Specific Program ENrichment (ASPEN), which incorporates two parallel methods that are simultaneously applied to a given dataset. First, expressed genes within each annotated cell type are ranked according to their strength of correlation with age, and then GSEA is performed (Fig. 2a). Second, using a signature scoring algorithm, each dataset is assigned an overall gene set score and the mean signature score per cell type per donor is correlated with age (Fig. 2b). Results are then visualized via a bubble plot (Fig. 2c). A pseudocode document describing ASPEN and an R script of the ASPEN framework is provided at https://github.com/adrienneparsons/BC_singlecell_age.

### Cell-specific age-related programs reveal global differences in TNBC and ER$^+$ breast cancer

Before implementing ASPEN, we established dataset selection criteria: sufficient sample sizes across different breast cancer subtypes, appropriate age ranges, inclusion of both tumor and stromal cells, and accepted cell-type annotations. The single-cell and spatially resolved human breast cancer atlas alone met these criteria[23]. The dataset includes ten TNBC samples (*n* = 42,512 total cells, mean age = 55.3 years, age range = 35–73) and 11 ER$^+$ samples (*n* = 38,241 total cells, mean age = 60.9 years, age range = 42–88). There were insufficient HER2$^+$ samples across the age spectrum for ASPEN analysis.

We first examined tumor compositions by assessing the contribution of eight major cell populations ('celltype_major': cancer epithelium, normal epithelium, cancer-associated fibroblasts (CAFs), myeloid cells, T cells, B cells, endothelium and perivascular-like (PVL) cells)[23] and found no age-associated changes in cell abundance in either TNBC or ER$^+$ breast cancer (Extended Data Figs. 2a and 3a). Analysis of 29 annotated subpopulations ('celltype_minor')[23] within major cell types showed no age-related differences in TNBC (Extended Data Fig. 2b). In ER$^+$ tumors, the proportions of inflammatory CAFs (iCAFs) decreased and myofibroblast-like CAFs (myCAFs) increased with age (Extended Data Fig. 3b).

We then applied ASPEN to identify age-related programs (ARPs), defined as the gene expression sets (for example, Hallmark pathways from the Molecular Signatures Database (MSigDB)) associated with age. Global analysis of normalized enrichment scores (NES) from the minor cell populations revealed that most ARPs were positively associated with age in TNBC and negatively associated with age in ER$^+$ breast cancer (Fig. 2d). Taken together, the results indicated that tumor-associated cell-type transcriptional programs, rather than their abundance, change with age.

Age-related, cell-type-specific enrichment patterns from ASPEN were unique to each breast cancer subtype (Fig. 3 and Extended Data Fig. 4a). Importantly, the ASPEN results (age correlation; Fig. 3) were concordant with the GSEA analyses of bulk transcriptomes (age stratification; Fig. 1) whereby immune programs increase with age in TNBC and epithelial/breast-cancer-associated programs decrease with age in ER$^+$ breast cancer. Applying ASPEN to 15 published senescence signatures revealed no consistent association with age in either TNBC or ER$^+$ breast cancer (Extended Data Fig. 4b).

### Cell-specific enrichments of EMT, immune responses and stress responses with age in TNBC

In TNBC, EMT in cancer epithelial subpopulations represented the strongest overall enrichment with increasing age (Fig. 3a and Source Data for Fig. 3). The EMT process confers enhanced tumor-initiating capacity, invasion and metastatic potential[24,25]. In basal cancer cells, enrichment of EMT coincided with increased immune response, KRAS signaling, apoptosis and angiogenesis, and decreased oxidative phosphorylation (OXPHOS), myc targets and E2F targets with age (Fig. 3a–e). Multiplexed immunofluorescence (mIF) on tumor tissue from an independent clinical cohort of young (<45) and older (>70) patients with TNBC confirmed age-associated EMT. While the extent of pan-cytokeratin (panCK) staining (to visualize epithelial cells) did not differ, co-staining for panCK and the mesenchymal protein, vimentin, was significantly higher in the older cohort (Fig. 4a,b and Source Data for Fig. 4).

Consistent with earlier analyses (Fig. 1c), immune function and inflammation were enriched with age in several cell populations (Fig. 3b and Source Data for Fig. 3). The most enriched immune ARPs, marked by the highest NES, were interferon (interferon-α (IFNα) and IFNγ) response pathways in both iCAFs and myCAFs (Fig. 3b and Source Data for Fig. 3). CD4$^+$ and CD8$^+$ T cells displayed elevated stress responses and apoptosis with age (Fig. 3d). Notably, monocyte and macrophage populations displayed no significant ARPs (Fig. 3a–e).

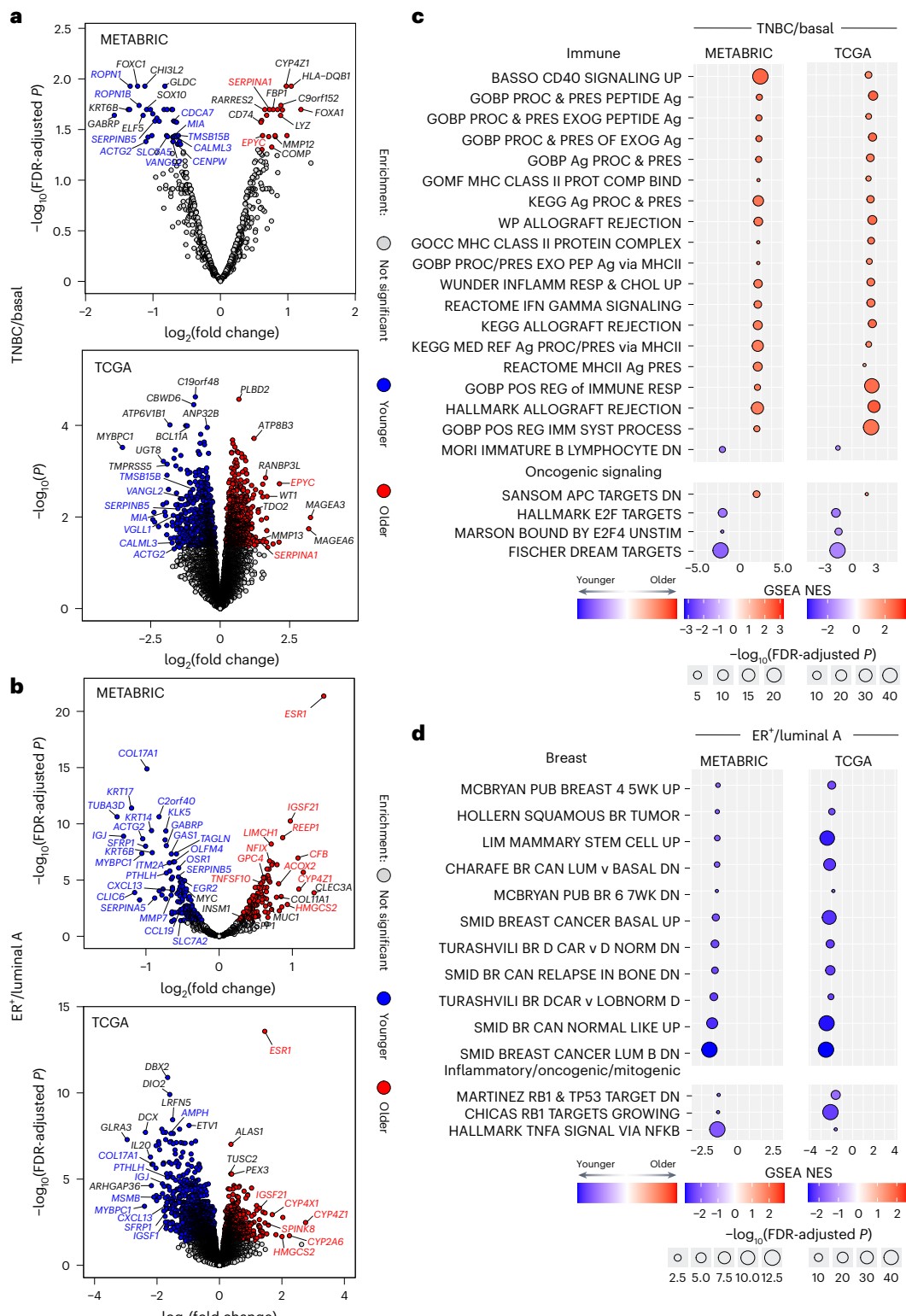

**Fig. 1 | Age-related DEGs and functional gene set enrichments in TNBC, basal, ER⁺ and luminal A breast cancers. a,b**, Volcano plots showing the log₂ fold change (*x*) and *P* value (*y*) based on gene expression analysis of tumors from patients with TNBC (METABRIC) and basal PAM50 subtype breast cancer (TCGA) (**a**) and ER⁺ breast cancer (METABRIC) and luminal A PAM50 subtype breast cancer (TCGA) (**b**), comparing the <45 years and >65 years age groups. The red dots represent genes enriched in the >65 age group; the blue dots are genes enriched in the <45 age group. Statistical significance was determined using an empirical Bayes-moderated two-sided *t*-test at a significance threshold of 0.05 for nominal *P* values (basal) or Benjamini–Hochberg-corrected *P* values (TNBC, ER⁺, luminal A). *n* = 50 TNBC < 45; *n* = 63 TNBC > 65; *n* = 86 ER⁺ < 45;

*n* = 386 ER⁺ > 65; *n* = 30 basal <45; *n* = 37 basal >65; *n* = 68 luminal A < 45; *n* = 152 luminal A > 45. **c,d**, Results of age-stratified GSEA of genes ranked according to the log₂ fold difference from **a** and **b** in TNBC/basal (**c**) and ER⁺/luminal A breast cancer (**d**). Statistical significance and normalized enrichment was determined using a permutation-based null distribution, per the calculations of the fgsea R package. Pathways are grouped according to biological similarity. The red fill color indicates enrichment in the >65 age group; blue indicates enrichment in the <45 age group. Circle size is proportional to relative −log₁₀(Benjamini–Hochberg $P_{adj}$) for the enrichment; color depth represents the magnitude of the normalized enrichment score (NES). Data for Fig. 1 are provided in Source Data for Fig. 1.

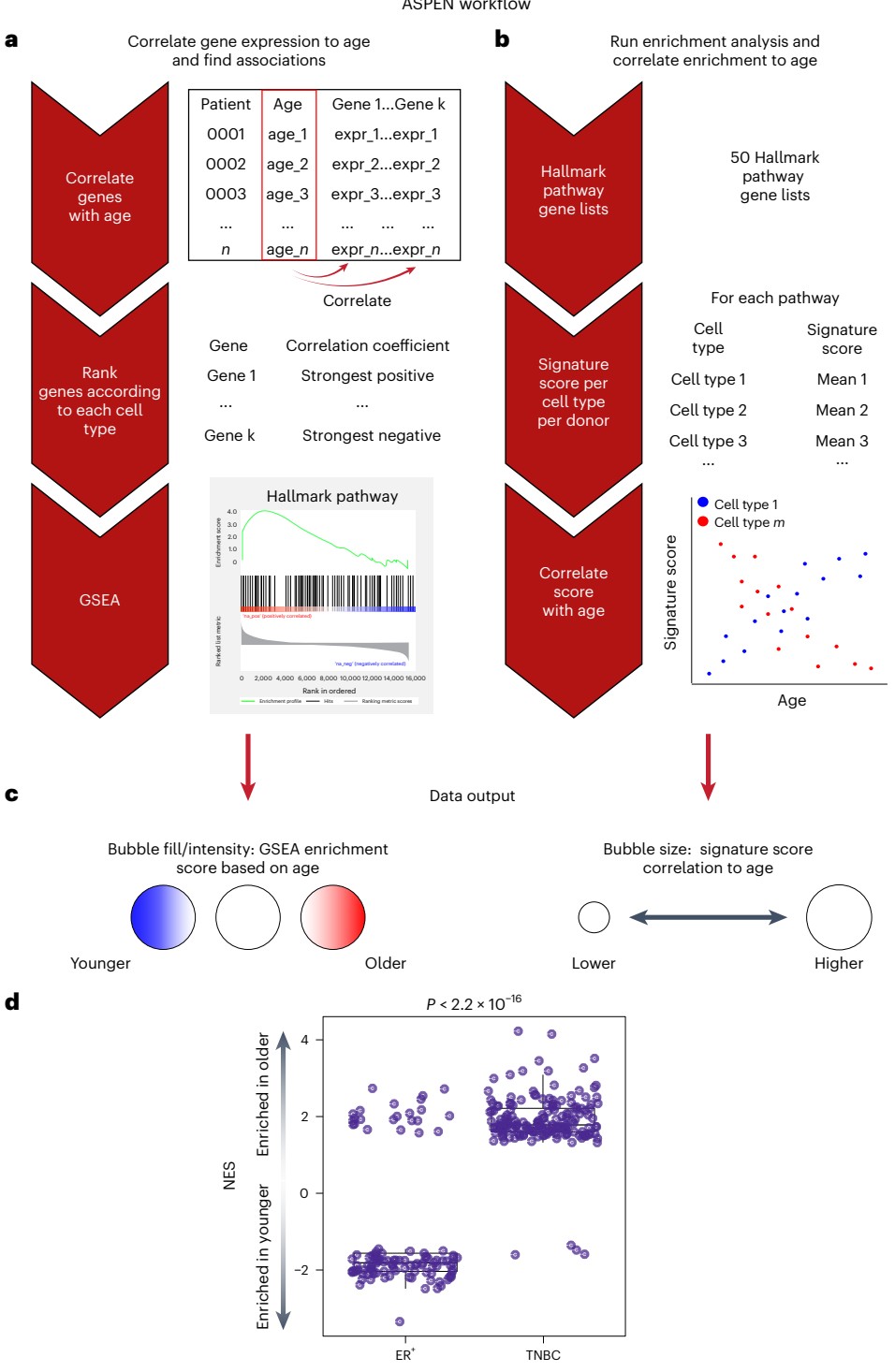

**Fig. 2 | Development of a single-cell ASPEN analysis pipeline and global enrichment methods.** ASPEN relies on parallel adaptations of GSEA and signature scoring to associate gene expression-based enrichment of functional pathways to age. **a**, The mean gene expression per cell type is matched to donor age and a correlation coefficient for each gene is calculated. The genes with nonzero coefficients are then ranked according to their correlation and GSEA is performed using select gene sets of choice (in our case, Hallmark). **b**, Concurrently, the gene sets are used to assign a signature score to every cell in the single-cell dataset using Seurat. After scoring, the mean signature score for each gene set is calculated per cell type per donor. These mean values are then correlated to donor age. **c**, The resulting NES from **a** are then plotted as the data point color for each cell type and pathway combination, with red indicating enrichment in older donors, blue indicating enrichment in younger

donors and white indicating a failure to achieve statistical significance (false discovery rate (FDR)-adjusted $P > 0.05$). Irrespective of the correlation direction (coefficient <0 or >0) in **b**, the magnitude of the correlation of signature score to age is visualized as the size of the data point for each cell type and pathway combination, with point size being proportional to the magnitude of correlation (larger circle = more strongly correlated or anticorrelated). **d**, Box plot showing the distribution of 175 TNBC and 110 ER⁺ NES values that achieved $P_{adj} < 0.05$ for each cell type and Hallmark pathway combination from ASPEN (colored circles in **c**; Fig. 3). An NES > 0 indicates significant enrichment in older patients; an NES < 0 indicates significant enrichment in younger patients. Significance was determined using a two-tailed Student's $t$-test on the NES for each breast cancer subtype. The center line indicates the median; the box limits indicate the upper and lower quartiles; the whiskers indicate 1.5 times the interquartile range.

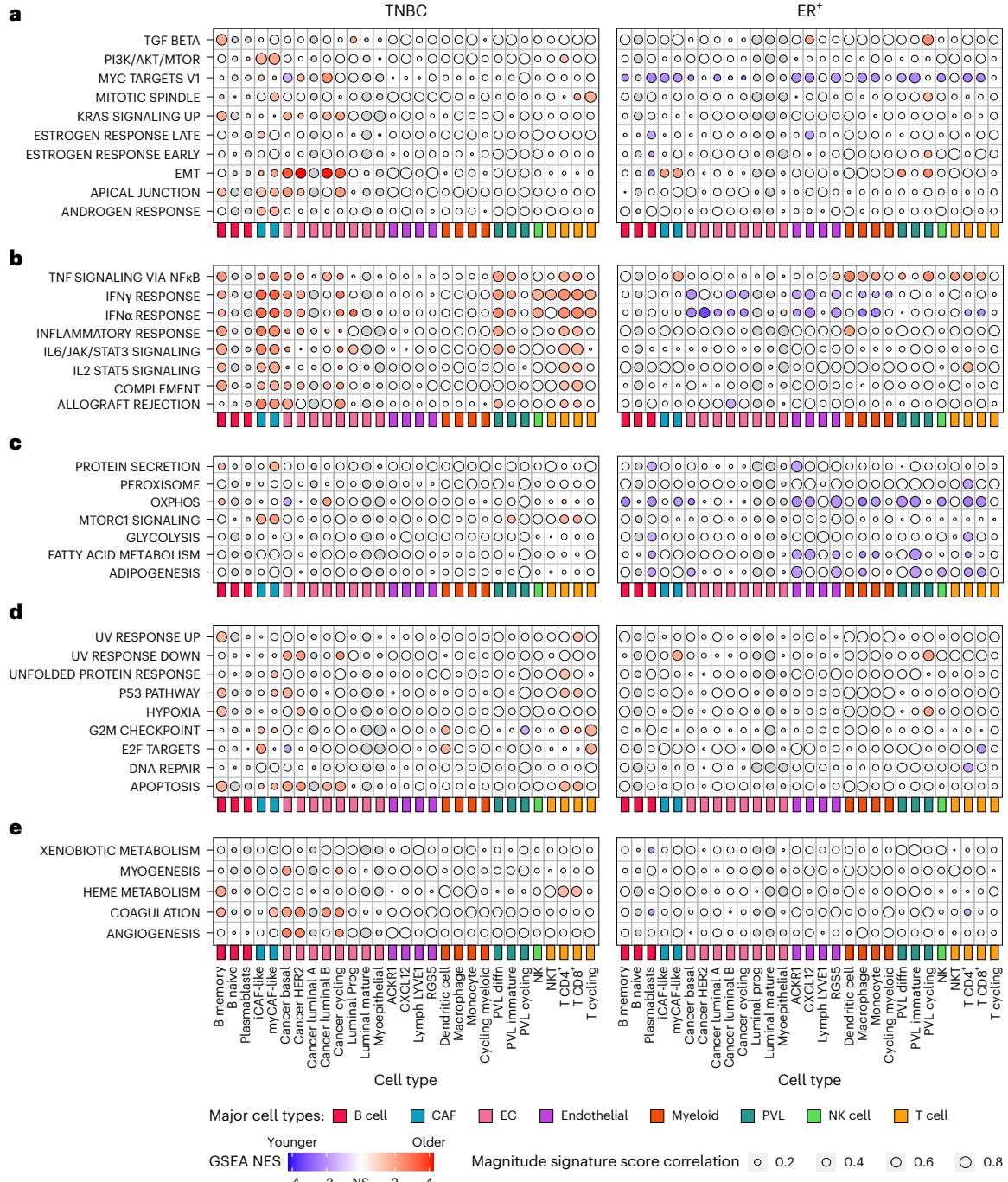

**Fig. 3 | Cell-specific ARPs in TNBC and ER⁺ breast cancer.** Results from the ASPEN analysis of the breast cancer scRNA-seq atlas dataset[23] and Hallmark gene sets yielding cell-specific ARPs in TNBC (left) and ER⁺ breast cancer (right). The bubble plot shows 29 minor cell types (color-coded according to major cell-type groups) on the *x* axes and Hallmark pathways on the *y* axes. ARPs were manually grouped into biologically similar processes: cancer-associated (**a**), immune-related (**b**), metabolism (**c**), cell stress/DNA repair (**d**) and others (**e**). Donors with a celfgseal count of zero for a given cell type were excluded from the analysis of that cell type. Statistical significance was determined at a threshold of $P_{adj} < 0.05$, whereby significance must be achieved in both the fgsea-derived permutation test and gage-derived two-sided Welch's *t*-test-

style parametric gene set test (Methods). Bubble color indicates the NES of the age-associated GSEA analysis (Fig. 2a), with a deeper color indicating greater enrichment. Red indicates significant enrichment in older donors; blue indicates significant enrichment in younger donors; white indicates no statistical significance; gray indicates cell types that were present in <50% of donors and were thus excluded from the analysis. Bubble size indicates the magnitude of the enrichment score correlation to age (Fig. 2b); larger bubbles indicate stronger correlation or anticorrelation. IFN, interferon; OXPHOS, oxidative phosphorylation; TGF, transforming growth factor. Data for Fig. 3 are provided in Source Data for Fig. 3.

These results established that with increasing age, TNBC is dominated by cancer cells with an EMT phenotype and an inflamed microenvironment in which T cells and CAFs display responses to cellular stress and immune stimuli.

## Cell-specific reductions in metabolism, myc targets and IFN responses with age in ER⁺ breast cancer

Consistent with the bulk analyses (Fig. 1d), most ER⁺ breast cancer ARPs were enriched in younger patients (Fig. 3, Extended Data Fig. 4

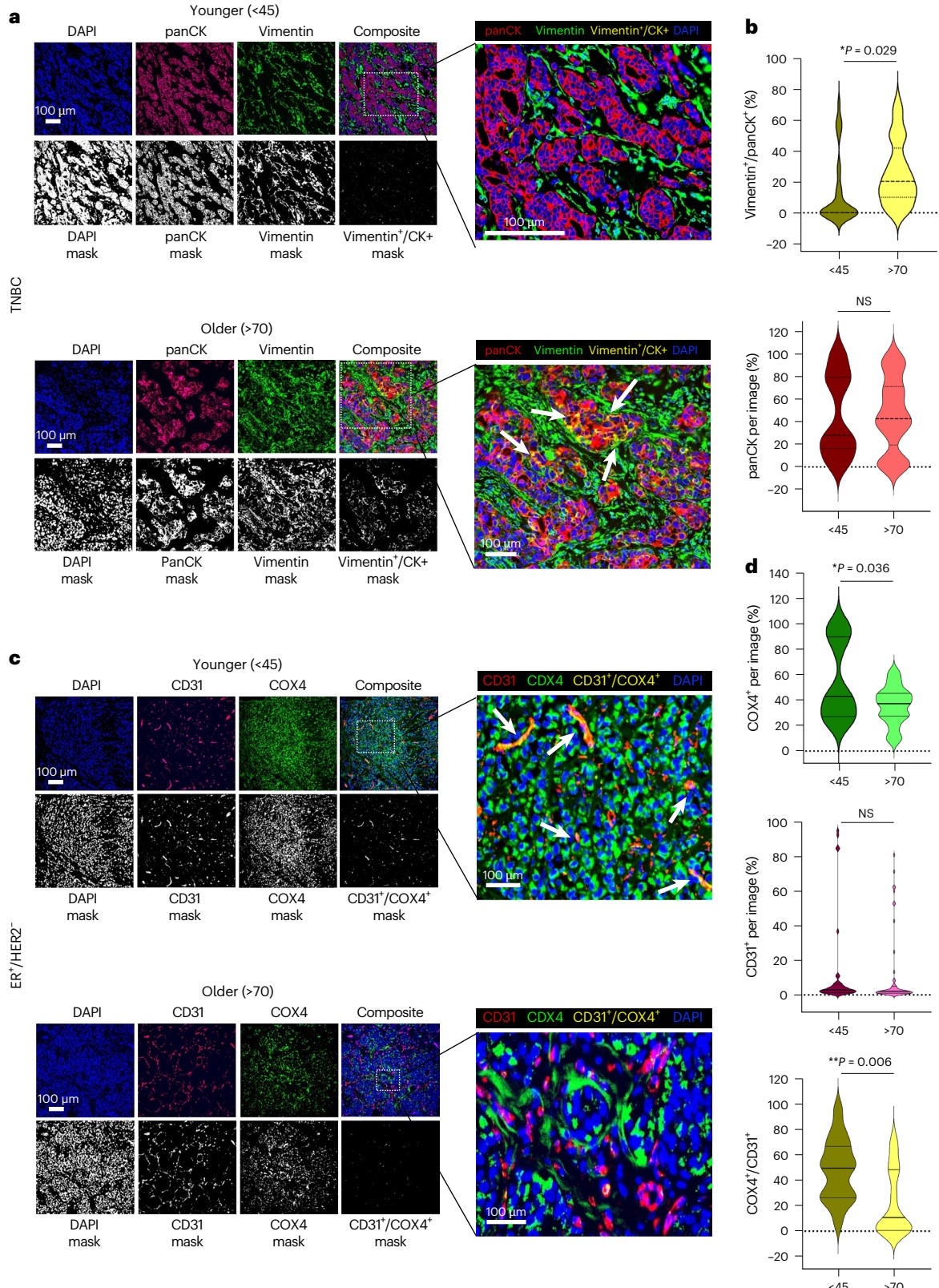

and Source Data for Fig. 3). Only IFN response and myc target ARPs were significantly enriched in epithelial subpopulations from the youngest donors (Fig. 3a,b and Source Data for Fig. 3). No estrogen response ARPs were detected in the tumor epithelium (Fig. 3a) despite high *ESR1* expression in tumors from older patients (Fig. 1b,d), further supporting the model of positive feedback regulation under low postmenopausal estrogen.

Metabolic processes were significantly enriched in tumors from younger donors with ER+, particularly in the vasculature, plasmablasts, and CD4+ and CD8+ T cells (Fig. 3c and Source Data for Fig. 3), which is consistent with reports of age-associated metabolic reprogramming of T cell immunity in cancer[26] and metabolic rewiring of vasculature[27,28]. Supporting these results, staining for COX4, a protein essential for oxidative phosphorylation, was significantly higher in an independent

**Fig. 4 | mIF analysis of tumors from younger and older patients with TNBC and ER⁺/HER2⁻ breast cancer. a**, Representative images of TNBC tumor tissue sections from younger (<45) and older (>70) patients stained for panCK (red) and vimentin (green). Nuclei are stained with 4′,6-diamidino-2-phenylindole (DAPI) (blue). For each age group, the top rows include individual fluorescent channels and the composite image; the bottom rows include the threshold masks for each channel and the overlap mask obtained for the green and red channels. Enlarged images represent composite images where colocalization of vimentin⁺/panCK⁺ cells is indicated in yellow; examples of colocalization are indicated by the white arrows. **b**, Violin plots representing the quantification of vimentin and panCK overlap staining as a percentage of total panCK staining per image (top) and total panCK staining per image (bottom) in each age cohort. Younger cohort n = 27 independent images representing five tumors; older cohort n = 41 independent images representing seven tumors. **c**, Representative images of ER⁺/HER2⁻ tumor tissue sections from younger (<45) and older (>70) patients stained for CD31 (red)

and COX4 (green). Nuclei are stained with DAPI (blue). For each age group, the top rows include individual fluorescent channels and the composite image; the bottom rows include the threshold masks for each channel and the overlap mask obtained for the green and red channels. Enlarged images represent composite images where colocalization of CD31⁺/COX4⁺ cells is indicated in yellow, and examples of colocalization are indicated by the white arrows. **d**, Violin plots representing quantification of COX4 staining per image (top), CD31 staining per image (middle) and CD31 and COX4 overlap as a percentage of total CD31 staining (bottom) in each age cohort. Younger cohort n = 29 independent images representing five tumors; older cohort n = 42 independent images representing seven tumors. Data for **b** and **d** are provided in Source Data for Fig. 4. Two-sided Wilcoxon rank-sum P values are indicated. Some of the violin plot boundaries exceed 100% or go below 0 because of kernel density smoothing; all values are within the 0–100% range. NS, not significant.

cohort of ER⁺ tumor tissue from younger (<45) compared to older patients (>70) (Fig. 4c,d and Source Data for Fig. 4). Moreover, while mean CD31⁺ endothelial staining was not significantly different between cohorts, the percentage of metabolically active (COX4⁺) endothelium was significantly higher in tumors from the younger cohort (Fig. 4d).

Myeloid cells in ER⁺ tumors were transcriptionally different with age. Specifically, TNF signaling was positively associated with age, while IFN responses showed negative associations with age (Fig. 3b and Source Data for Fig. 3). Monocytes and macrophages were also less metabolically active with age (Fig. 3c). These results suggested type I inflammatory responses in ER⁺ tumors from younger patients and a tumor-promoting inflammatory phenotype[29] in the older cohort.

ASPEN also revealed age-related differences in T cell populations. Specifically, the TNF and interleukin-2 (IL-2) signaling pathways increased while IFNα responses, myc targets and metabolism pathways decreased with age in CD4⁺ and CD8⁺ T cells (Fig. 3a–c).

Overall, ER⁺ breast cancer ARPs indicated reduced metabolic activity across many cell types and particularly the endothelium, enrichment of tumor-supportive inflammatory activity in myeloid cells and attenuated IFN responses in cancer cells with age. The ARPs in CD4⁺ and CD8⁺ T cells suggested quiescence, exhaustion and metabolic dysfunction with increasing age.

### Age-differential cellular interactomes and spatial relationships in TNBC and ER⁺ breast cancer

We next investigated age-specific cell–cell interactions using Cell-Chat, which integrates single-cell transcriptomic expression of

ligands, receptors, cofactors, multimeric receptor–ligand complexes, soluble agonists and antagonists, and stimulatory and inhibitory membrane-bound co-receptors, along with cell type abundance, to infer the likelihood of ligand–receptor interaction between cell types[30]. To achieve balanced group sizes and statistical robustness, we stratified patients in the breast cancer atlas dataset[23] into cohorts aged 55 years and younger and 55 years and older for both subtypes (Source Data for Fig. 5).

In TNBC, the older cohort exhibited a 1.85-fold increase in total cell–cell interactions and a 1.48-fold increase in interaction strength (Fig. 5a,b and Source Data for Fig. 5). Both younger and older cohorts with TNBC showed strong interactions between T cells and cancer epithelial cells, as well as homotypic T cell interactions (Fig. 5a,b). The older cohort's interactome was dominated by bidirectional myeloid-to-T cell communication, while the younger cohort's tumors displayed higher CAF interaction probabilities with T cells and cancer epithelium (Fig. 5c and Source Data for Fig. 5). These results align with the METABRIC, TCGA and ASPEN results, highlighting age-related immune program enrichments (Figs. 1a,c and 3b).

In ER⁺ breast cancer, the older cohort had a 1.16-fold increase in total interactions but a 1.06-fold decrease in interaction strength (Fig. 5d,e and Source Data for Fig. 5). Both age groups displayed strong CAF–cancer epithelial interactions (Fig. 5d,e). Interaction probabilities between cancer epithelium and both myeloid and T cells were elevated in the older cohort, while those within the vasculature—endothelium and PVLs—were enriched in the younger cohort (Fig. 5f), which is consistent with the increased vascular metabolic activity observed through ASPEN and mIF staining (Figs. 3c and 4c,d).

**Fig. 5 | Age-related cell–cell interactions in TNBC and ER⁺ breast cancer. a–f**, Circos plot showing the predicted homotypic and heterotypic interaction strength between major cell types in TNBC (**a–c**) and ER⁺ breast cancer (**d–f**) tumors from the scRNA-seq atlas[23] using the CellChat analysis. Circos plots are shown for younger patients (≤55, **a,d**), older patients (>55, **b,e**) and the differential between age groups (**c,f**). TNBC ≤ 55 years (n = 6, n = 20,591 cells), TNBC > 55 years (n = 4, n = 20,203 cells), ER⁺ ≤ 55 years (n = 6, n = 21,735 cells), ER⁺ > 55 years (n = 5, n = 15,344 cells). The indicated cell types are represented by colored nodes. **a,b,d,e**, The edge colors correspond to the source cell type. **c,f**, The edge colors indicate stronger interaction strength between cells in either the older (red) or younger (blue) patient tumors. Edge thickness is proportional to the strength of interaction between the given cell types. **a,b,d,e**, Information included with the ordinate labels indicates the total number of interactions (I) and total interaction strength (S) for each cohort. **g,j**, Heatmaps representing differential interaction strengths between each indicated target (x axes) and source (y axes) cell for TNBC (**g**) and ER⁺ breast cancer (**j**). The color scale is based on the differential interaction strength; shades of red indicate stronger interaction in the older cohort; shades of blue are stronger in the younger cohort. For example, homotypic macrophage interactions are stronger in older patients with TNBC compared to younger patients. The bar plots at the top of the heatmaps correspond to the absolute sum of differential incoming interaction strength for each cell type; the bar plots to the right of the heatmaps correspond

to the absolute sum of differential outgoing interaction strength for each cell type. Cell-type color annotations are consistent throughout (**g,j**). **h,i,k,l**, mIF imaging and quantification of tumor microarrays of patients with TNBC and ER⁺. Each tissue core was imaged in its entirety and 1–3 cores were analyzed per patient. Representative images of patients aged 55 years and younger and older than 55 years show staining for panCK (red), CD8 (pink), α-smooth muscle actin (SMA) (white), CD31 (orange), CD140 (green) and nuclei (blue); autofluorescence (AF) appears gray. For TNBC, the box plots represent the median percentage of CD8⁺ T cells (pink) located within 30 µm of the tumor epithelium (red) (**h**); the median percentage of CD8⁺ T cells (pink) located within 30 µm of CAFs (CD140b⁺/CD31⁻/SMA⁺/⁻, green or white) (**i**); n = 94 patients ≤55 years; n = 127 patients >55 years. For ER⁺ breast cancer, the box plots represent the median percentage of CD8⁺ T cells (pink) located within 30 µm of the tumor epithelium (red) (**k**), and the median percentage of CD31⁺ ECs (orange) located within 30 µm of CAFs (CD140b⁺/CD31⁻/SMA⁺/⁻, green or white) (**l**); n = 132 patients ≤55 years; n = 237 patients >55 years. Statistical significance was determined using a two-sided Wilcoxon rank-sum test, adjusted using the Benjamini–Hochberg method. For the box plots, the data points represent the median values for a single patient; the center line represents the median; the box limits represent the upper and lower quartiles; and the whiskers represent 1.5 times the interquartile range. **h,i,k,l**, Data are provided in Source Data for Fig. 5.

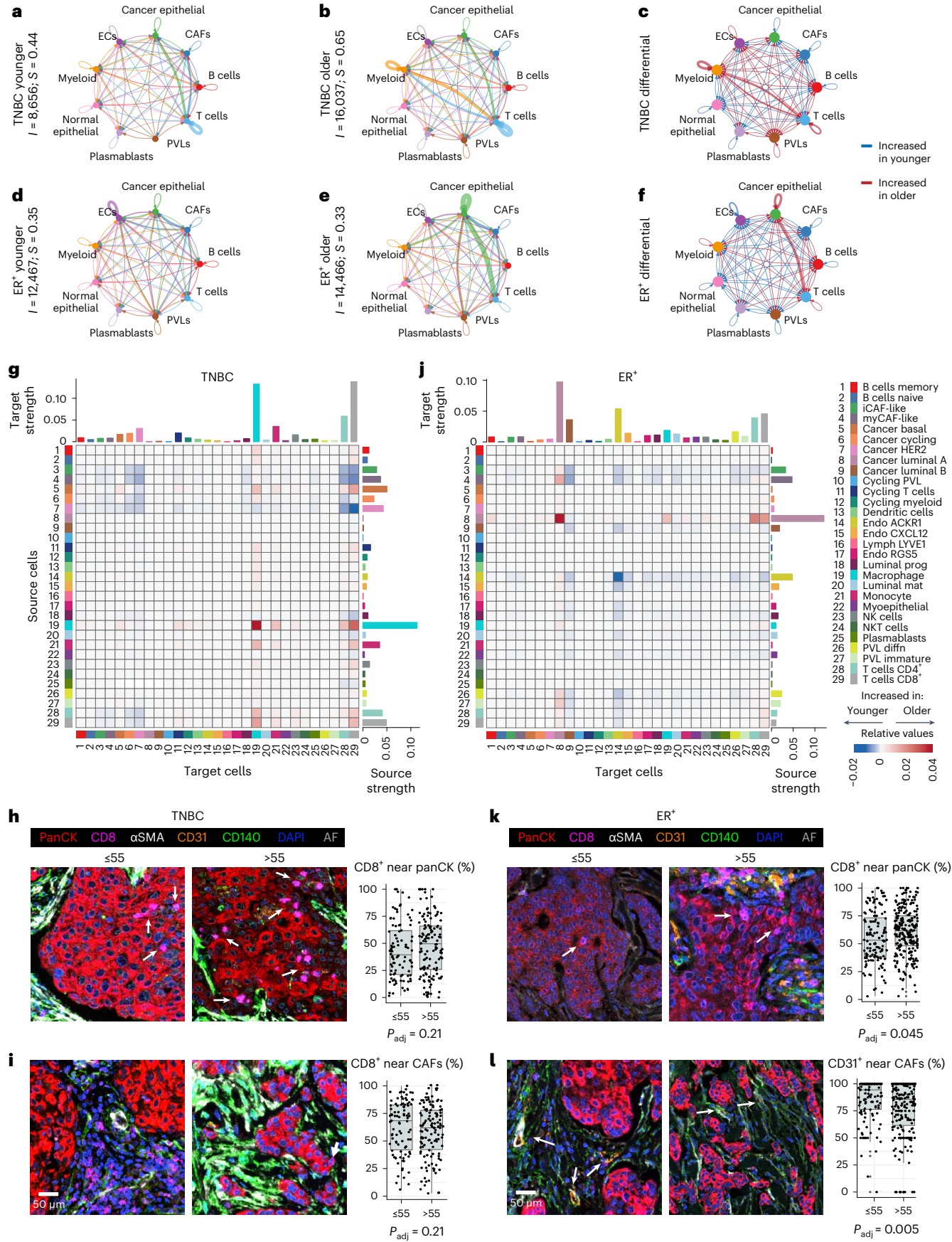

To ensure that interactions were not predicted from cells of different patients, we calculated the communication probabilities for each patient sample separately. For both disease subtypes, results of the patient-specific analysis were consistent with the global age cohort aggregates (Extended Data Fig. 5a,b).

These results prompted us to identify cells that accounted for the most robust age-related interactions. Examining the 29 minor cell subpopulations revealed many age-stratified interactions (Source Data for Fig. 5). We describe only the most predominant below.

In the older cohort with TNBC, cancer basal cells showed the strongest predicted interactions with CD8[+] T cells, CD4[+] T cells and macrophages (Fig. 5g). To further investigate the proximity of CD8[+] T cells to cancer cells, we performed spatial mIF analysis on a TNBC tissue microarray (TMA) built from an independent patient cohort using the same age stratification (≤55, >55). In the older cohort, 49.2% of CD8[+] T cells were located within 30 μm of at least one tumor cell, a threshold selected based on both cell adjacency and the effective range of most cytokines[31]. Conversely, 42.6% of CD8[+] T cells were within this range in the younger cohort (Fig. 5h and Source Data for Fig. 5). Although this difference did not reach statistical significance, a key consideration is that panCK labels all epithelial cells in the mIF analysis, whereas single-cell RNA sequencing (scRNA-seq) resolves specific epithelial subpopulations that may or may not display age-associated interactions with CD8[+] T cells. Hence, the observed trends confirm that a greater proportion of CD8[+] T cells are within functional communication distance of tumor cells in the older cohort.

In TNBC, although CAF-specific ARPs increased with age (Fig. 3), CAF interaction probabilities were higher in the younger cohort, particularly for myCAF and iCAF signaling to CD4[+] and CD8[+] T cells (Fig. 5g). Spatial mIF analysis of the independent clinical cohort confirmed more CD8[+] T cells in the younger group (68.3%) were located within 30 μm of CAFs than in the older group (61.3%), although this observation did not reach statistical significance (Fig. 5i and Source Data for Fig. 5).

Also in TNBC, macrophages exhibited the strongest age-related changes, including more homotypic interactions and increased communication with cancer basal cells, monocytes, CD4[+] T cells and CD8[+] T cells in tumors from older patients (Fig. 5g), despite their lack of ARPs (Fig. 3). The increased macrophage–T cell interactions and transcriptomic signals of enhanced MHC class II presentation with age (Fig. 1c) led us to examine cell-specific MHC class II expression. While professional antigen-presenting cells (APCs) accounted for most of the human leukocyte antigen (HLA) gene expression (encoding MHC class II molecules), age-biased expression was driven by CAFs, vascular cells and cancer cells in older patients (Extended Data Fig. 6a), suggesting IFNγ exposure[32] and aligning with enriched IFN response genes in these cells (Fig. 3b).

In ER[+] breast cancer, luminal A cancer cells exhibited the most dramatic age-related interaction changes, significantly increasing autocrine and immune cell interaction probabilities with age (Fig. 5j and Source Data for Fig. 5). Spatial mIF analysis of an independent ER[+] clinical cohort TMA using the same age stratification (≤55, >55) confirmed that significantly more CD8[+] T cells were localized within 30 μm of cancer epithelial cells in tumors from older patients than in those from younger patients (Fig. 5k and Source Data for Fig. 5). Prompted by the enhanced activity of the ER[+] breast cancer epithelium and elevated *ESR1* with age (Fig. 1c), we assessed cell-specific *ESR1* expression and found that it was significantly higher in luminal A and B cancer cells in older patients than those in younger patients (Extended Data Fig. 6b).

Also in ER[+] breast cancer, atypical chemokine receptor (*ACKR1*)[+] endothelial cells (ECs) had 15-fold increased homotypic interaction probabilities and more predicted interactions with several cell populations in the younger cohort (Fig. 5j), aligning with their enhanced protein secretion and metabolic activity ARPs in younger patients (Fig. 3c). The *ACKR1* gene product modulates innate immunity by trafficking chemokines[33]; thus, these results—coupled with the observed

EC IFN response ARPs—suggest stronger immune modulation in the younger cohort. *ACKR1*[+] ECs also had stronger interaction probabilities with both iCAFs and myCAFs in the younger cohort (Fig. 5j). Analysis of tissue from an independent cohort confirmed that the proportion of ECs within 30 μm of CAFs was significantly higher in tumors from younger patients (Fig. 5l and Source Data for Fig. 5).

### Identifying age-stratified signaling networks in TNBC and ER[+] breast cancer

We next explored the molecular basis for age-biased cell–cell interactions in TNBC and ER[+] breast cancer. In the CellChat database, specific ligand–receptor pairs are categorized into general signaling pathways[30]. We use 'signaling interaction' to denote signaling pathways predicted to be activated between specific cell types, and 'signaling node' for the ligand–receptor pair(s) activated between cells. To avoid producing an overwhelming number of signaling nodes, we developed regression-based selection criteria to prioritize the most prominent cell subsets and their signaling interactions (Methods). Results for each breast cancer subtype are described in the following sections.

### Age-associated signaling in TNBC

The selection criteria yielded seven cell types for TNBC: iCAF, myCAF, basal cancer cells, macrophages, monocytes, CD4[+] and CD8[+] T cells. We used the CellChat rankNet function to calculate scaled interaction weights across the 49 possible source and target combinations, yielding 650 signaling interactions including 71 different signaling pathways; 483 (74%) of these had higher probability values in the older cohort, 307 signaling interactions were exclusive to the older cohort and 48 were exclusive to the younger cohort (Supplementary Figs. 1–3). Nine signaling pathways, supported by 43 ligand–receptor pairs, were the most dominant across all selected cells in one or both TNBC age cohorts (Fig. 6a, Supplementary Figs. 1–3 and Source Data for Fig. 6).

The monocyte/macrophage-derived GALECTIN signaling interaction showed the greatest difference between TNBC age groups and was elevated in older patients (Fig. 6a and Source Data for Fig. 6). Notably, signaling nodes between cancer basal cells expressing *P4HB* (encoding prolyl 4-hydroxylase beta polypeptide) and both monocyte/macrophages and CAFs expressing *LGALS9* (encoding galectin-9) were exclusive to the older cohort (Fig. 6a, Supplementary Fig. 4 and Source Data for Fig. 6). Galectin-9 signaling via P4HB is linked to EMT promotion and age-related cancers[34]. Moreover, the plasminogen activator-urokinase (*PLAU*) and cyclophilin A (*PPIA*) signaling nodes, which promote EMT in tumor cells and fibrosis in CAFs[35–37], were enriched between cancer epithelial cells (receptors) and both monocyte/macrophage and CAFs (ligands) in the older cohort (Fig. 6a and Supplementary Fig. 4). These results suggested that myeloid cells and CAFs provide a source of EMT-promoting factors in an age-dependent manner and may help explain the enrichment of EMT we observed in older patients (Figs. 3a and 4a,b).

As observed repeatedly, immune modulatory signaling nodes were more prominent in the older cohort with TNBC. First, type 2 IFN (IFNγ) signaling interactions from CD8[+] T cells to iCAFs, myCAFs and cancer basal cells were exclusive to the older cohort (Supplementary Figs. 3b and 4, and Source Data for Fig. 6), supporting our earlier findings (Figs. 1 and 3). Second, in the older cohort, myeloid cells and CAFs were enriched for immunosuppressive phenotypes, most notably *LGALS9* signaling to CD4[+] and CD8[+] T cells via *CD44* (Fig. 6a and Supplementary Fig. 4). Galectin-9-CD44 signaling promotes CD8[+] T cell death[38], aligning with the ASPEN results establishing increased enrichment of a T cell apoptosis pathway with age (Fig. 3d). Third, predicted interaction between *CD55* expressed by CAFs, basal cancer cells and monocyte/macrophages and the adhesion G protein-coupled receptor E5 (CD97) gene, *ADGRE5*, on CD4[+] T cells was higher with age (Fig. 6a and Supplementary Fig. 4). In CD4[+] T cells, galectin-9-CD44 and CD55-CD97 signaling promote regulatory T (T_reg) cell function[38]

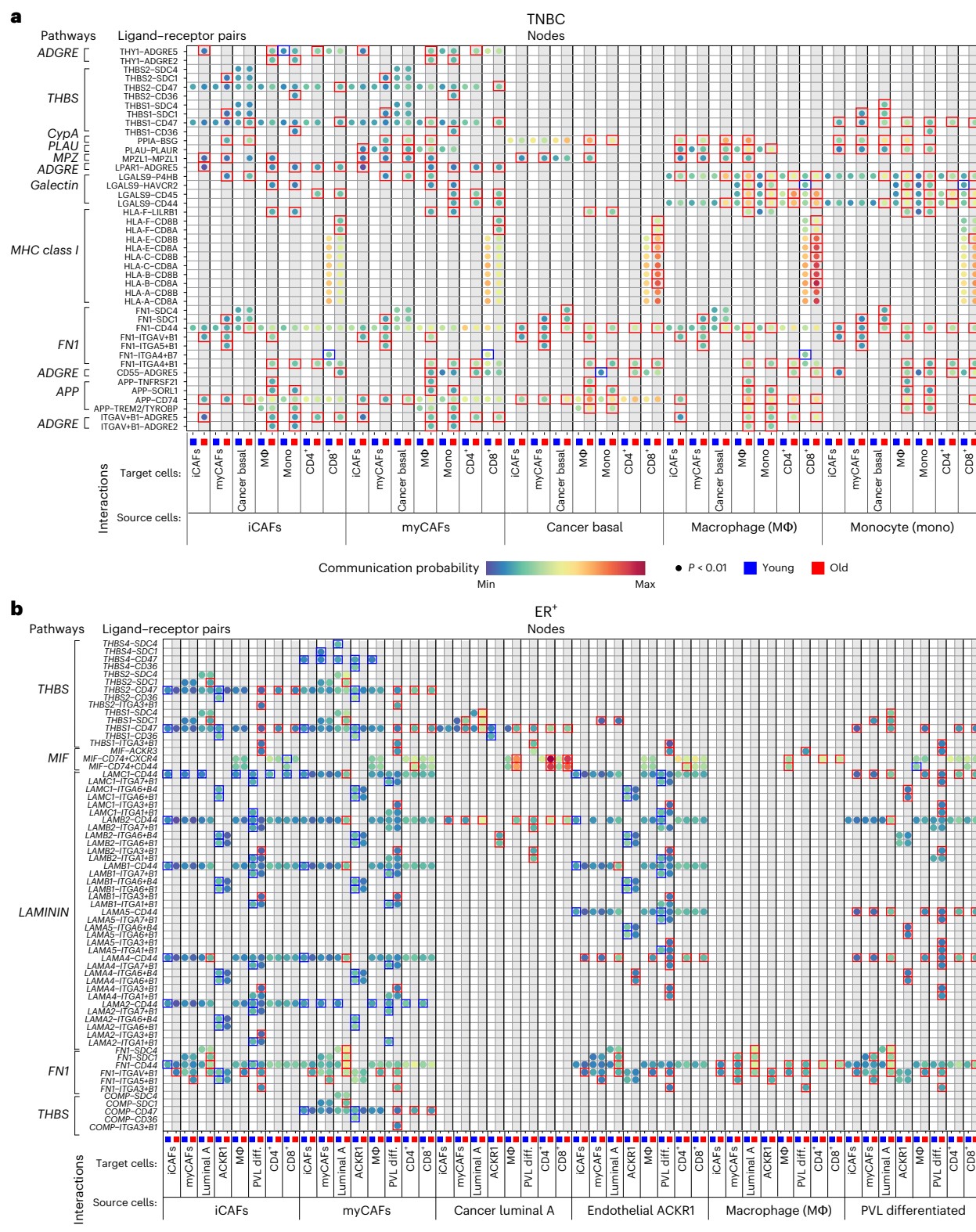

**Fig. 6 | Age-associated signaling nodes. a,b,** Bubble plots representing the communication probability in TNBC (**a**) and ER⁺ breast cancer (**b**) for each indicated ligand–receptor pair between the indicated source and target cells for each age cohort (Methods, Source Data for Fig. 6 and Supplementary Figs. 1–8). The rows depict the ligand–receptor pairs and signaling pathways; the columns depict specific source–target cell interactions for the ≤55 cohort (blue) adjacent to the >55 cohort (red). Communication probabilities are represented by a color scale from deep blue (minimum) to green, yellow, orange and deep red (maximum). Each bubble represents a signaling node predicted to be active through a nonparametric permutation test with *P* < 0.01 through CellChat probability calculations[30]. The colored boxes around bubbles indicate signaling nodes with higher communication probabilities in the younger (blue boxes) or older (red boxes) cohort (criteria: *P* < 0.01 in only one age group or in both age groups with a fold difference of ≥1.2). diff., differentiated.

and development[39], respectively, suggesting enhanced T_reg-mediated immunosuppression with age. We observed additional immunosuppressive factors (for example, prostaglandin E2, *PGE2*, and macrophage migration inhibitory factor, *MIF*) expressed by myeloid cells in the older cohort's tumors (Supplementary Figs. 2b, 3a and 4, and Source Data for Fig. 6).

Although signaling nodes involving *CD8* on CD8+ T cells and MHC class I-related genes on cancer basal cells were elevated in the older cohort (Fig. 6a and Supplementary Fig. 4), cancer cell-derived immunosuppressive signaling interactions were also increased. For example, immunosuppressive interactions involving the *PGE2*, *MIF* and midkine (*MK*) gene products were more likely between cancer basal cells and CD4+ and CD8+ T cells in the older cohort (Supplementary Figs. 2a and 4). Immunosuppressive signaling nodes[40,41] between the MHC class I molecule, *HLA-F*, on both cancer basal cells and CAFs and the inhibitory receptor, leukocyte immunoglobulin like receptor B1 (*LILRB1*), on monocyte/macrophages, were also noted in older patients (Fig. 6a and Supplementary Fig. 4).

Signaling nodes related to MHC class II presentation by cancer basal cells, myCAFs and iCAFs to monocytes and macrophages were elevated in the older cohort (Supplementary Figs. 1–4), confirming our earlier observation of increased expression of various MHC class II-related genes in 'nonprofessional' APCs with age (Extended Data Fig. 6a).

In CAF populations, signaling interactions involving lysophosphatidic acid receptor 1 (*LPAR1*), thymus cell antigen 1 (*THY1*, encoding CD90) and thrombospondins 1 and 2 (*THBS1, THBS2*) were prominent in the older cohort (Fig. 6a and Supplementary Fig. 4). *LPAR1* signaling via *ADGRE5* promotes fibrosis and chemoresistance in TNBC[42]. *THY1*+ CAFs, annotated as iCAFs in our dataset[23], suppress T cell function[43], promote T_reg recruitment[44] and are associated with poor outcome in glioblastoma[45]. We also observed elevated *THBS1* signaling probability, particularly via the gene that encodes syndecan-1 (*SDC1*), in CAFs in the older cohort (Fig. 6a and Supplementary Fig. 4). Thrombospondin 1 aids cancer cell motility[46] and is associated with reduced breast cancer survival[47]. These results suggest that CAFs, which highly express immune and inflammatory response ARPs (Fig. 3b), also express factors that promote cancer progression with age.

Few of the signaling interactions were higher in the younger cohort with TNBC. A signaling node between monocyte/macrophage-derived *LGALS9* and CD8+ T cell *HAVCR2* (encoding the TIM3 checkpoint protein that suppresses antitumor immunity[48] and induces CD8+ T cell death[49]), was elevated in the younger cohort (Fig. 6a and Supplementary Fig. 4). The high probability of macrophage-derived osteopontin (secreted phosphoprotein 1, *SPP1*) signaling to various cells exclusively in the younger cohort (Supplementary Figs. 2b and 4) suggests promotion of breast cancer progression and chemoresistance[50–53]. Higher probabilities of fibronectin 1 (*FN1*) engagement of integrins α4/β7 (*ITGA4* and *ITGB7*) on CD8+ T cells suggested regulation of T cell migration and T cell receptor activity in the younger cohort (Fig. 6a and Supplementary Fig. 4). Finally, CAF signaling to lymphocytes and myeloid cells via *MIF* occurred exclusively in the younger cohort (Fig. 6a, and Supplementary Figs. 1a,b and 4). Hence, while certain signaling pathways like GALECTIN and MIF were broadly activated across the TME in the older cohort, they were also implicated in specific interactions in the younger cohort.

### Age-associated signaling in ER+ breast cancer

For ER+ breast cancer, the selection criteria revealed eight cell types: iCAF, myCAF, cancer luminal A, macrophages, *ACKR1*+ ECs, differentiated PVLs, CD4+ T cells and CD8+ T cells. Across 64 possible combinations, rankNet analysis yielded 745 signaling interactions consisting of 84 signaling pathways (Supplementary Figs. 5–8 and Source Data for Fig. 6). Of those interactions, 411 (55%) were more prevalent in the younger cohort, while 334 were more prevalent in the older cohort. The younger cohort had 166 unique interactions while 186 were unique to

the older cohort (Supplementary Figs. 5–8 and Source Data for Fig. 6). Four signaling interactions (FN1, laminin, MIF and THBS), supported by 64 ligand–receptor pairs, were identified as the most dominant across all selected cells with age (Fig. 6b, Supplementary Figs. 5–8 and Source Data for Fig. 6).

Factors involved in adhesion and extracellular matrix (ECM) engagement included most of the strong signaling interaction probabilities across selected cell types in both ER+ breast cancer age cohorts. These included gene products of the non-collagenous glycoprotein family–laminins, fibronectin and thrombospondin–interacting with various heterodimeric integrin receptor gene products (Fig. 6b, Supplementary Figs. 5–8 and Source Data for Fig. 6). ECM engagement within the TME modulates cell proliferation, differentiation, adhesion and migration, serves as a sink for cytokines, promotes angiogenesis and inflammation, and governs malignant progression[54]. Adhesion-related signaling nodes were both age-dependent and cell-type-specific; nearly all adhesion/ECM signaling nodes involving either cancer luminal A cells, differentiated PVLs or macrophages were enriched in the older cohort, while those involving iCAFs, myCAFs and *ACKR1*+ ECs were mostly enriched in the younger cohort (Fig. 6b). T cells showed no engagement of these particular adhesion molecules (Extended Data Fig. 7).

Examination of the most robust interactions, specifically those involving *ACKR1*+ ECs and PVLs, yielded signaling nodes underlying the vascular ARPs and spatial relationship between CAFs and ECs that we had observed (Figs. 3–5). For example, the PERIOSTIN signaling interaction, which mediates fibrosis, angiogenesis and chemoresistance in cancer[55], was highly enriched between CAFs and ECs/PVLs in the older cohort (Supplementary Figs. 5–9 and Source Data for Fig. 6). Our analysis also predicted age-biased signaling in vascular cells through integrins and laminin subunits, which are the major non-collagenous components of the basement membrane[54]. For example, the predictions suggested that *ACKR1*+ ECs preferentially use laminin subunit alpha 4 (*LAMA4*) to engage other cells in tumors from the older cohort, while using other laminins to engage those same cells in the younger cohort (Fig. 6b and Supplementary Fig. 9). Likewise, based on *ITGA3* and *ITGB1* expression in differentiated PVLs, increased signaling through the alpha-3/beta-1 integrin was predicted in the older cohort (Fig. 6b and Supplementary Fig. 9).

Immunosuppression with age was also evident in ER+ breast cancer. Predicted interactions between *MIF*-expressing luminal A cancer cells and CD4+ or CD8+ T cells expressing CD74 complex genes (*CD74* and *CXCR4*; *CD74* and *CD44*) were elevated in the older cohort (Fig. 6b, Supplementary Figs. 5–9 and Source Data for Fig. 6). Tumor cell-derived MIF inhibits CD8+ T cell activation and promotes expansion of T_reg cells[56], which is consistent with the IL-2 ARP we observed in CD4+ T cells (Fig. 3b). Predicted signaling via *SPP1*-expressing macrophages, which promotes disease progression, was exclusive to tumors from the older cohort (Supplementary Figs. 5–8); this is consistent with the METABRIC analysis, which revealed significantly higher expression of *SPP1* in older patients' tumors (Fig. 1b).

Finally, our analysis predicted enrichment of Notch signaling from PVL cells and myCAFs to cancer luminal A cells in the younger cohort (Supplementary Figs. 6b, 7a and 9, and Source Data for Fig. 6). Notch has a critical role in maintaining luminal progenitor cell fate in the breast[57], supporting the luminal and stem cell pathway enrichments observed in the younger METABRIC cohort (Fig. 1d). While overexpression of Notch receptors and ligands is correlated with TNBC progression and therapeutic resistance, it is less well described in ER+ breast cancer[57].

### Integrated models of the age-related landscapes of TNBC and ER+ breast cancer

Through integration of key cell-specific results from computational and spatial analyses, we built comprehensive age-related landscapes of TNBC and ER+ breast cancer (Supplementary Fig. 10).

Our results support a TNBC model (Supplementary Fig. 10a) whereby in older cohorts, myeloid cells and CAFs interact with cancer basal cells via LGALS9/P4HB, PPIA/BSG and PLAU/PLAUR signaling to promote EMT and cell motility. Mesenchymal-like cancer cells in turn affect the TME by (1) presenting antigen to T cells; (2) promoting T_reg development; (3) inducing CD8$^+$ T cell death; and (4) generating an immune suppressive phenotype in monocyte/macrophages. Older patients' tumor cells were also predicted to engage with CAFs to promote fibrosis and ECM remodeling, which is required for mesenchymal-like tumor cells to detach, thereby enhancing motility, invasion, metastasis and chemoresistance. CAF gene expression programs indicated their dominant role in modulating immune responses, as evidenced by their inflammatory ARPs and signaling nodes that suppress T cell function and recruit T_reg cells. Increased MHC class II-related gene expression in many cells within the TME of older patients indicates IFNγ exposure, supported by elevated IFN response ARPs, particularly in CAFs. Given the dual role of IFNγ in promoting antitumor immunity and mediating immune evasion[58,59], IFN signaling in CAFs warrants further investigation, particularly in the context of aging. The absence of monocyte/macrophage ARPs in TNBC aligns with our finding that these cells use different signaling nodes in older versus younger cohorts to achieve tumor promotion and immunosuppression, which are linked to TNBC progression and metastasis[60], suggesting that age-stratified strategies are required to target tumor-associated macrophages.

An ER$^+$ breast cancer model (Supplementary Fig. 10b) is supported by evidence of increased myeloid inflammatory activity, less metabolically active endothelium, attenuated cancer cell IFN responses, and CD4$^+$/CD8$^+$ T cell quiescence and metabolic dysfunction with age. Those processes, which also impinge on cell migration, vascular permeability and immune trafficking, are influenced by ECM structure and alignment[61]. Although the specific signaling nodes were dissimilar, both age cohorts expressed signaling nodes involved in adhesion and ECM interactions, suggesting age-biased tissue remodeling as a key driver of cell-specific ARPs. The fact that *ACKR1*$^+$ ECs, which are involved in chemokine trafficking in innate immunity, were the most metabolically active cells in the younger ER$^+$ cohort, suggests an important immunomodulatory role within the younger ER$^+$ TME. Results from the older cohort suggest that CAFs promote increased desmoplasia, chemoresistance and cancer cell invasion. Perhaps explaining, at least in part, why ER$^+$ tumors are generally immunologically 'cold'[62], the older cohort showed increased MIF pathway activity, elevated CD47 signaling and heightened TNF signaling. These age-related changes may attenuate T cell cytotoxicity and reduced immunogenicity through heightened inflammation[63,64].

## Discussion

We provide an age-resolved human breast cancer landscape by defining transcriptomes, interactomes, spatial relationships and signaling pathway activity for TNBC and ER$^+$ breast cancer at cell-type-specific resolution. The integrated approach we applied reveals molecular and cellular profiles that differentiate tumors from older and younger patients with breast cancer in a subtype-dependent manner with important implications. Our main findings suggest that age-associated EMT and pro-tumorigenic signaling networks may confer the poor outcomes typically experienced by older patients with TNBC. Given the role of EMT in metastasis, fibrosis and therapeutic resistance, targeting EMT-related pathways, or the age-related signaling networks that induce them, may be particularly relevant for improving outcomes in older patients with TNBC. In ER$^+$ breast cancer, age-related metabolic decline and immune dysfunction may contribute to differences in disease progression and treatment responses, suggesting a need for therapies targeting immune activation, vascular remodeling or metabolic support for older patients with this disease subtype. Different therapeutic approaches, such as those designed to target

high metabolic activity in immunosuppressive myeloid cells, may be required for younger patients with ER$^+$ breast cancer.

The results also underscore the risks of generalizing aging effects, reinforcing that aging varies not only across tissues and cancers, but also within specific cancer subtypes[65,66]. For example, increased EMT with age has been observed in pan-cancer bulk analyses[67], but our results suggest increased EMT only in older patients with TNBC. One likely explanation is that EMT capacity and other tumor cell-specific, age-related differences between subtypes are due to differences in the cell of origin. However, it is also important to consider the striking subtype-specific differences in age-dependent stromal and immune cell profiles and their impact on tumor cells.

Further investigation into the therapeutic implications of age-specific and subtype-specific immune responses in different cell types is warranted. It is unclear why immunotherapies have limited efficacy in TNBC. The enrichment of immunosuppressive pathways in the older cohort with TNBC suggests a lower threshold for overcoming immunosuppression in younger patients. Findings like the strong immunomodulatory impact of CAFs suggest ways to investigate intrinsic and acquired resistance in TNBC. In ER$^+$ breast cancer, age-stratified therapies have been proposed[11,68]. Tissue and vascular remodeling, which affect immune infiltration and drug accessibility, also emerged as key age differentiators in ER$^+$ breast cancer, encouraging age-related therapeutic design.

A key strength of our study is the analysis of multiple independent clinical cohorts, spanning both transcriptomic and proteomic modalities. This approach enabled consistent, well-supported insights into how age at diagnosis shapes the breast TME, providing a robust resource for hypothesis generation and further investigation. Limitations of our study include lack of outcome data and the inability to uncouple age and menopause in our breast cancer datasets. Given that menopause is a feature of aging, it is reasonable to hypothesize that some of our observations are driven by menopausal status. As single-cell atlases with clinical outcome data become available, opportunities to study age-defined disease progression and therapeutic response will arise. While our focus was on breast cancer, analyses of normal breast tissue datasets[69,70] using similar approaches may provide insight into age-related cancer initiation. For example, age-related changes in breast biology may influence the distribution of breast cancer molecular subtypes[71–73]. Studies of aging normal breast tissue revealed key hallmarks—immune functional decline and loss of lineage fidelity[74–76]—that enable cancer development.

The ASPEN framework is extensible beyond breast cancer to any tissue from which scRNA-seq data are available. ASPEN implements common methods, like Pearson correlation, signature scoring and GSEA, to uncover age-related changes in gene expression in two ways: (1) correlating gene expression to age to generate a ranked output for GSEA; and (2) using signature scoring-based correlation of pathway enrichment with age. This dual evaluation increases confidence in findings. ASPEN could thus correlate any continuous variable to gene expression to resolve transcriptional enrichments at cell-type resolution, provided there are sufficient sample sizes in the data.

Our study establishes that the breast TME differs profoundly with age in a subtype-specific manner, suggesting that efforts to affect breast cancer pathology, design efficacious therapies and improve patient outcomes should consider subtype-related aging hallmarks.

## Methods

### Research cohort ethics and approval

All publicly available and validation data were derived from human, female patients with breast cancer. We included experimental validation through mIF of tumor sections and TMAs. Participants were not compensated for inclusion in this study.

Tissue samples consisted of surgically resected, formalin-fixed, paraffin-embedded breast tissue sections from older (>70 years,

grade I–III disease) and younger (<45 years, grade I–III disease) patients. Samples from the older cohort were provided in a de-identified manner under Mass General Brigham institutional review board approval no. 2021P001031. Tissue collection was performed with institutional review board approval from all participating institutions and according to the Declaration of Helsinki prior to the 2024 revision. All patients provided written informed consent. Tumor tissue was analyzed from patients with ER$^+$/HER2$^-$ ($n = 6$) and triple-negative ($n = 8$) breast cancer from the ELEVATE (ClinicalTrials.gov registration: NCT03818087) and ADVANCE (ClinicalTrials.gov registration: NCT03858322) studies. ER$^+$/HER2$^-$ and TNBC tissues for the younger cohorts were procured from AMSBio.

ER$^+$ TMA samples were from the Breast Boost cohort recruited to the St George Breast Boost study between 1998 and 2003 (Clinical Trials registration: NCT00138814). The TNBC TMA cohort consists of TNBC cases diagnosed between 2004 and 2019 at St George Hospital, Sydney, Australia (not collected under a clinical trial). Ethics approval was granted by the South Eastern Sydney Local Health District Human Research Ethics Committee at the Prince of Wales Hospital, Sydney (Boost: HREC 96/16 and TNBC: HREC 2018/ETH00138) who granted a waiver of consent to perform research analyses on the tissue blocks. All methods were performed in accordance with the relevant institutional guidelines and regulations. The raw clinical data are not publicly available because of ethics restrictions.

### DEGs in younger and older patients with breast cancer

Stage I–III donors in the METABRIC bulk RNA expression database were grouped into ER$^+$ ('ER$^+$ high prolif' or 'ER$^+$ low prolif') or TNBC ('ER$^-$/HER2$^-$') and according to age (<45 years and >65 years). Median age for the >65 TNBC group was 70.6 years (65.16–96.29 years, $n = 63$). The <45 TNBC group had a median age of 39.8 years (26.72–44.93 years, $n = 50$). Median age for the >65 ER$^+$ group was 72.8 years (65.02–92.14 years, $n = 386$). The <45 ER$^+$ group had a median age of 41.1 years (26.26–44.99 years, $n = 86$).

Donors with stage I–III breast cancer from the TCGA bulk RNA expression database were grouped into luminal A or basal molecular subtypes and according to age (<45 years and >65 years). Median age for the >65 basal group was 72.0 years (66–90 years, $n = 37$). The <45 basal group had a median age of 40.0 years (29–44 years, $n = 30$). Median age for the >65 luminal A group was 73.5 years (66–90 years $n = 152$). The <45 luminal A group had a median age of 39.0 years (26–44 years $n = 68$).

Log$_2$-normalized METABRIC gene expression data, as well as batch-normalized gene expression from TCGA Illumina HiSeq_RNASeqV2 and all associated metadata, were downloaded from the cBioPortal. For METABRIC, for each breast cancer subtype, genes ($n = 24,174$ total genes), were first subsetted to the top 675 genes with the highest variance by ranking genes according to standard deviation. Data were then transformed by exponentiating by base 2. For the TCGA, for each breast cancer subtype, we increased our statistical power by filtering low-expressed genes, keeping genes whose count per million value exceeded 1 in at least ten samples. We continued the analysis with 16,151 genes in ER$^+$ and 15,439 genes in TNBC for the TCGA.

For both METABRIC and TCGA, the limma (v.3.60.6) R package was used to calculate normalization factors alongside voom to identify DEGs between donors <45 years and >65 years in the TNBC/basal and ER$^+$/luminal A cohorts per dataset. We set the Benjamini–Hochberg-adjusted $P$ value significance threshold for both cohorts in METABRIC, and the luminal A cohort in the TCGA to 0.05. Adjusting $P$ values for the basal TCGA cohort eliminated all but one significant DEG, so for downstream GSEA analysis we used unadjusted $P$ values, with a threshold of $P < 0.05$. All 675 high-variance genes in METABRIC were then visualized in a volcano plot ($x = \log_2$ fold change, $y = -\log_{10}$(Benjamini–Hochberg-adjusted $P$)). For the TCGA, all 16,151 genes in the ER$^+$ cohort analysis were visualized in a volcano plot ($x = \log_2$ fold change, $y = -\log_{10}$(Benjamini–Hochberg-adjusted $P$)), while all 15,439 genes in the TNBC cohort analysis were visualized in a volcano plot with $x = \log_2$ fold change and $y = -\log_{10}(P)$.

The total list of genes for each subtype and dataset was ranked according to $\log_2$ fold change, starting with highest positive (most enriched in >65) and ending with lowest negative (most enriched in <45). GSEA was performed using the fgsea package (v.1.30.0) on the C2, C5 and Hallmark pathway gene sets (MSigDB). Common pathways with Benjamini–Hochberg-adjusted $P < 0.05$ for TNBC/basal and ER$^+$/luminal A were then visualized.

### Cell composition analyses

scRNA-seq counts matrices, barcodes, feature data and metadata for 10 TNBC and 11 ER$^+$ primary breast tumors were downloaded from the Gene Expression Omnibus (GEO) (GSE176078); metadata were downloaded from the accompanying paper's supplementary data. Using the Seurat R package (v.5)[77] to identify age-associated trends in cell type composition, we calculated the proportion of each cell type of middle granularity ('celltype_minor') as a proportion of its corresponding major cell type for each donor and correlated the proportions to donor age. $P$ values were corrected using the Benjamini–Hochberg method. The proportion of minor cell types within each major cell type for each donor, and the correlations for each minor cell type to age, were visualized.

### ASPEN determines Hallmark pathway association with age at cell-type resolution in scRNA-seq data

We developed ASPEN (Age-Specific activation Program ENrichment) to assess the relationship between gene set (pathway) expression and biological variables, such as age, at cell-type resolution. We used age as a continuous biological correlate with gene expression in every annotated cell type detected in TNBC and ER$^+$ tumors in the breast cancer atlas. A Seurat object was made for each of the 21 samples in the single-cell atlas and the ten TNBC donors' objects and 11 ER$^+$ donors' objects were merged into a single TNBC object and a single ER$^+$ object. From there, data were log-normalized and we assessed the correlation between donor age and Hallmark pathway enrichment per cell type as described below. The middle-granularity cell-type annotations provided by the authors of the dataset were used (29 total cell types, celltype_minor).

For the first arm of ASPEN, the TNBC or ER$^+$ merged object was resubsetted according to donor. Within each of the ten (TNBC) or 11 (ER$^+$) objects, the mean gene expression for each cell type was calculated per gene. Some donors had a cell count of zero for specific cell types; these donors were excluded from the next steps of the analysis. Cell types present in less than half of the donors were also excluded. The mean expression values for each gene for that cell type were correlated to donor age (a total of 24 unique cell types for TNBC and 25 unique cell types for ER$^+$). Each gene per cell type was then ranked from highest correlation coefficient (most correlated) to lowest (most anticorrelated). Genes with a correlation coefficient of zero were omitted; the remaining ranked genes were used to perform GSEA for the Hallmark pathways. GSEA was performed using the fgsea (v.1.30.0) and gage (v.2.54.0) R packages; this portion of the script was adapted from a publicly available GSEA script developed by B. Gudenas (https://bioinformaticsbreakdown.com/how-to-gsea/). Gene set .gmt files were accessed from the GSEA website. For a given cell type and pathway combination to be considered a statistically significant enrichment, $P_{adj} < 0.05$ in both package analyses was required (see 'Statistical analysis' section).

For the second arm of ASPEN, the AddModuleScore command in Seurat v.4 was used to assign a signature score to each cell in the TNBC or ER$^+$ merged object for gene expression concordance with the 50 Hallmark pathways. These gene sets were accessed in R using the msigdbr (v.25.1.1) R package. Once every cell per disease subtype had a signature score, the objects were subsetted according to donor and the

mean signature score per cell type was calculated. The resulting mean signature score per cell type per donor was correlated to donor age. The outputs of both arms were then visualized.

## mIF analysis for EMT and oxidative phosphorylation

Tissue mIF cohort sample sizes included $n = 6$ TNBC > 70, $n = 5$ TNBC < 45, $n = 7$ ER$^+$ > 70 and $n = 5$ ER$^+$ < 45. All tissues were processed and analyzed in blinded fashion. Briefly, 5-μm sections were deparaffinized and rehydrated. The slides were incubated with antigen retrieval buffer (10 mM citric acid, pH 6.0, +0.05% Tween-20) for 10 min at 100 °C in a microwave with a reduced power setting. All slides were blocked in 5% normal goat serum (NGS) for 1 h at room temperature. Then, slides were incubated at 4 °C overnight with the following primary antibodies diluted in 5% NGS: pan-cytokeratin (1:100, clone AE1/AE3, host: mouse, cat. no. NBP2-24949, Novus Biologicals), vimentin (1:50, clone SP20, host: rabbit, cat. no. MA5-16409, Invitrogen), CD31 (1:50, clone RM247, host: rabbit, cat. no. MA5-33063, Invitrogen), COX4 (1:50, clone 4D11-B3-E8, host: mouse, cat. no. 11967S, Cell Signaling Technology). Slides were then washed with PBST (1× PBS + 0.1% Tween-20). Detection was performed using secondary antibodies: Donkey anti-Rabbit IgG (H+L) Highly Cross-Adsorbed Secondary Antibody, Alexa Fluor 488 (cat. no. A-21206, Invitrogen); Goat anti-Mouse IgG (H+L) Highly Cross-Adsorbed Secondary Antibody, Alexa Fluor 594 (cat. no. A-21145, Invitrogen); Goat anti-Mouse IgG (H+L) Highly Cross-Adsorbed Secondary Antibody, Alexa Fluor 647 (cat. no. A-21240, Invitrogen); and Goat anti-Rabbit IgG (H+L) Highly Cross-Adsorbed Secondary Antibody, Alexa Fluor 647 (cat. no. A-21244, Invitrogen) at a dilution of 1:750 in 0.1% NGS for 1 h at room temperature. After washing the slides in PBST, a Vector TrueVIEW Autofluorescence Quenching Kit (cat. no. SP-8400-15, VectorLabs) was used to quench autofluorescence from the tissues. Nuclei were stained with DAPI (cat. no. D1306, Invitrogen) and the sections were mounted using ProLong Gold Antifade Mountant (cat. no. P36930, Invitrogen). Images were acquired on a Nikon Eclipse 90i microscope at ×20 magnification; 5–9 images were analyzed per tumor. Image analysis was performed using Fiji (v.2.9.0)[78]. For each image, individual channels were first separated and then converted to grayscale. A threshold was applied using the automated Otsu method in the Adjust Threshold tool. The image calculator function was then used to calculate pixel counts for each individual channel and the overlap between channels.

## Cell–cell interaction analysis

For the analysis of cell–cell interactions, we used CellChat (v.2.1.2)[30]. The single-cell human breast cancer atlas data were divided into four groups according to subtype and patient age at diagnosis (Source Data for Fig. 5). To define cohorts, we set our thresholds to ≤55 (young) and >55 (aged) to distribute sample sizes evenly across both subtypes. We created eight CellChat objects, consisting of cell-type annotations at two granularities (celltype_major and celltype_minor) and associated data, aggregating the scRNA-seq data for TNBC donors ≤55, TNBC donors >55, ER$^+$ donors ≤55 and ER$^+$ donors >55 before CellChat object creation and signaling probability calculations. To consider cell composition when calculating the interaction probability, the population.size argument in the computeCommunProb function was set to TRUE. For each subtype, we merged the CellChat objects to compare young and aged groups. The analysis determined the number and strength of interactions between cell types in the different cohorts and visualized them using tools like netVisual_circle and netVisual_diffInteraction to show the overall interaction probabilities between cell types of interest within or across age groups, rankNet to describe statistically significant, pathway-specific differences between a given source and target cell group between donors ≤55 and >55, or netVisual_bubble, which calculates the communication probability of each ligand–receptor interaction between source and target cells in each age cohort for a given pathway or pathways. Fold change differences above 1.2 or

below −1.2 in the bubble plot comparisons were of interest and were further indicated.

## Regression-based criteria for curating cells and signaling nodes

Cell types of interest were chosen based on source or target signal strengths greater than the sum of the mean source/target interaction strengths of the younger and older cohorts (>0.037 for TNBC and >0.023 for ER$^+$ breast cancer; Source Data for Fig. 5). The criteria yielded seven cell types for TNBC. For ER$^+$ breast cancer, these criteria revealed eight cell types. We excluded three cell types despite meeting the criteria: (1) TNBC cancer HER2 cells because only one patient sample contained appreciable numbers of cancer HER2 cells (Extended Data Fig. 2a); (2) TNBC cycling cancer ECs because we did not know their intrinsic molecular subtype; and (3) ER$^+$ breast cancer luminal B cells because luminal A cells had a 6.7-fold higher interactome enrichment with age (Source Data for Fig. 5).

We then applied the rankNet function to these seven or eight cell types as both source and target cells (49 total interactions for TNBC and 64 total interactions for ER$^+$) to identify the signaling pathways through which these cell types were interacting. The probabilities of specific ligand–receptor interactions ('signaling nodes') for each signaling pathway category identified in the rankNet analysis were extracted for further investigation.

To select the most relevant pathways for the downstream analysis, we then performed a univariate logistic regression model with the glm function of the stats R package. We averaged the interaction probabilities for each ligand–receptor pair in each pathway, source cell and target cell interaction for the >55 or <55 age groups. These means were used as input to the glm function, where each pathway was the test variable and age group was the response variable. We then selected pathways that had a $P < 0.05$ and appeared at least 15 times in the 49 or 64 interactions analyzed. We used the netVisual_bubble function to compare the upregulated and downregulated ligand–receptor pairs between age groups, setting a threshold $P < 0.01$ to highlight the ligand–receptor pair interactions with the highest confidence. For these, we identified signaling interactions that were either exclusive to one age group or exhibited fold change differences above 1.2 or below −1.2 between age groups to identify the most robust differences.

In some cases, where indicated, we nominated additional select signaling nodes by manual curation of significant rankNet interaction pathways that are (1) known to modulate the cell states and phenotypes that we observed in the METABRIC dataset or ASPEN analyses, (2) evidence-based age-related factors or (3) highly age-biased but restricted to fewer than 15 cell–cell interactions.

## Single-donor versus aggregated CellChat analysis

To ensure that the CellChat analyses were not skewed by our decision to aggregate data from each subtype and age group, we predicted cell–cell interaction strengths of the major cell types after creating a CellChat object for each donor. We then examined the predicted interaction strengths for the major cell types that include the minor cell types deemed of interest from the regression-based selection for each individual donor and compared those interaction strengths to the corresponding signaling strengths calculated by aggregating the data for all TNBC or ER$^+$ donors ≤55 or >55, as described above.

## TMA sample preparation and cell proximity mIF (for validation of major cell–cell interactions)

The TMA analysis cohorts included sample sizes of $n = 127$ TNBC > 55, $n = 94$ TNBC ≤ 55, $n = 237$ ER$^+$ > 55 and $n = 264$ ER$^+$ ≤ 55. mIF was carried out by staff at the Katharina Gaus Light Microscopy Facility, University of New South Wales. Tissue slides were initially baked at 58 °C for 60 min, then underwent dewaxing using xylene for 2 × 5 min, then

100% ethanol for 3 × 1 min, 70% ethanol for 1 × 1 min and finally distilled water for 1 × 1 min. This was followed by a 10-min wash in distilled water.

To determine optimal staining conditions, all antibodies were first tested using 3,3′-diaminobenzidine detection (BOND Polymer Refine detection, cat. no. DS9800, Leica Biosystems). Antigen retrieval was performed sequentially using citrate and then EDTA-based antigen retrieving buffers at 110 °C for 5 min. The staining process was automated using the Leica BOND RX system (Leica Biosystems). Immunohistochemical staining was used as a control to compare with mIF staining. Staining was performed using a panel of eight primary antibodies to identify stromal and immune subsets: panCK (1:2,000 dilution, clone AE1/AE3, host: mouse, cat. no. ab27988, Abcam); PDGFRβ (CD140b, 1:1,000 dilution, clone Y92, host: rabbit, cat. no. ab32570, Abcam); αSMA (1:500 dilution, polyclonal, host: rabbit, cat. no. ab5694, Abcam); CD146 (1:1,250 dilution, clone EPR3208, host: rabbit, cat. no. ab75769, Abcam); THY1 (CD90, 1:4,000 dilution, clone EPR3133, host: rabbit, cat. no. ab133350, Abcam); CD8 (1:1,000 dilution, clone C8/144B, host: mouse, cat. no. MA5-13473, Invitrogen); PD-1 (1:50 dilution, clone EPR4877(2), host: rabbit, cat. no. ab137132, Abcam); and CD31 (1:100 dilution, clone JC70A, host: mouse, cat. no. M0823, Agilent Technologies), with DAPI as a nuclear counterstain. These markers were selected based on prior single-cell analysis of TNBC.

The mIF imaging was conducted using Opal 9 (Akoya Biosciences) with primary antibody conditions optimized based on prior 3,3′-diaminobenzidine staining results for both single and multiplex assays. Each antibody was assigned a specific Opal fluorophore (OPAL650 for PanCK, OPAL540 for PDGFRβ, OPAL780 for αSMA, OPAL570 for CD146, OPAL690 for THY1, OPAL620 for CD8, OPAL520 for PD-1 and OPAL480 for CD31), considering biomarker co-expression and expected protein expression levels. Biomarkers located within the same cellular compartment were paired with spectrally distinct Opal fluorophores. Staining intensity and quality were assessed before proceeding with multiplexing, with DAPI as a nuclear counterstain. Normal tissue cores were included as internal controls to ensure consistent staining intensity across all slides. During the multiplex optimization phase, antibody concentrations were further refined to standardize signal intensity where needed. Each core was imaged once; mean cores analyzed for the ≤55 ER+ cohort: 2.5 (1–3 per patient); mean cores analyzed for the >55 ER+ cohort: 2.27 (1–3 per patient); mean cores analyzed for the ≤55 TNBC cohort: 2.28 (1–3 per patient); mean cores analyzed for the >55 TNBC cohort: 2.13 (1–3 per patient).

Images were preprocessed in QuPath v.0.2.3. PanCK was used to segment the tumor epithelium and stroma. DAPI was used for cell segmentation using QuPath's built-in algorithm. Individual cell classifiers were generated for each antibody, defining cell types according to marker combinations: CD8+ T cells (CD8+), iCAFs (CD140b+), myCAFs (αSMA+CD146+), ECs (CD31+ or CD31+CD146+) and tumor cells (panCK+). To account for potential misclassified cells, we excluded myCAFs, iCAFs and ECs identified within tumor regions, and tumor cells detected within stromal regions, from the spatial analysis. TMA cores with <10% or >90% stromal region were excluded because of insufficient data, often associated with artifacts or missing morphology.

### Spatial analysis of TMA mIF

The information of cells derived from TMA cores, including coordinates, marker intensity, cell types and tumor or stromal region assignments, was organized into a SPIAT object to facilitate spatial analysis (SPIAT v.1.8.0)[79]. CD8+ T cell proximity to epithelial cells or CAFs was measured using frNN (dbscan v.1.2.0, R version v.4.4.2), with cells within 30 μm considered adjacent. The same method quantified ECs near tumor. TMA core metrics were summarized at the patient level using the median.

### Statistical analysis

For the computational analyses of the scRNA-seq atlas, we excluded HER2+ samples (n = 5) because there were insufficient sample numbers

and age ranges for ASPEN and the downstream analyses. Therefore, for the bulk transcriptomic analyses of METABRIC and TCGA, we also excluded HER2+ patients, and patients with late-stage or stage 0 disease. Before the analysis, we also decided to exclude METABRIC and TCGA patients aged 45–65 years to align with established clinical risk. In some ASPEN and CellChat analyses, the exclusion criteria were preestablished and some cell types were not analyzed because of insufficient representation of the cell type across patients, as described above.

All experiments and analyses were performed with biological replicates. Sample sizes for computational methods relied on the availability of publicly available datasets used in the analysis. Sample sizes for experimental validation relied on patient enrollment and sample availability. No statistical methods were used to determine sample sizes. Sample sizes are outlined in the source data and in the text. All available data were analyzed, barring any exclusions described above. Samples were analyzed based on established disease subtypes and age ranges; all available tissue was analyzed per cohort, barring any exclusions as described above. Therefore, randomization was not necessary for this study. Tissue-based assays and analyses were performed in blinded fashion. For the computational analysis of existing data, blinding was not performed because it was necessary to classify samples into known subtype and age categories.

All analyses were performed in the R programming language and implemented statistical tests specific to the R package or algorithm applied when appropriate. All assumptions of statistical tests were met for each analysis. Where indicated, P values were corrected using the Bejamini–Hochberg method.

Analysis of DEGs in METABRIC and TCGA was performed using a standard pipeline in limma with voom normalization; DEGs were identified by a Benjamini–Hochberg-adjusted P value at a threshold of P < 0.05 for the ER+ METABRIC, luminal A TCGA and TNBC METABRIC analyses. For the basal TCGA data, we show the unadjusted P < 0.05. GSEA enrichment used Benjamini–Hochberg correction and a significance threshold of P < 0.05 to identify enriched pathways.

The cor.test function in R was used to calculate the relationship between cell-type proportions and age. As each minor cell-type proportion was calculated as a subset of its major cell population, Benjamini–Hochberg correction was applied to minor cell types within a major group, with a significance threshold of 0.05.

ASPEN uses the fgsea and gage BioConductor packages for significance testing. For each cell type, FDR correction at a significance threshold of P < 0.05 was used for both fgsea (using Benjamini–Hochberg correction) and gage (using default FDR calculations) across all analyzed pathways. A pathway was only considered significant if it reached the adjusted P value threshold for both packages.

CellChat analyses relied on probability calculations specific to the CellChat package. The rankNet function uses permutation testing to determine significant pathway communication differences between age groups. Univariate logistic regression for determining pathways of interest implemented a Wald test for the P value calculation for each pathway. We selected pathways of interest as those with P < 0.05 across at least 15 different source cell and target cell combinations, as described above. No further P value adjustment was made on the results of the univariate analyses; it was unnecessary because the P value threshold was simply implemented as a signifier that a pathway should be explored further, and no further biological conclusions were made from the logistic regressions. In the ligand–receptor visualizations, we relied on the CellChat algorithm's probability calculation to identify the P values of the communication probabilities. We set the threshold for visualization at 0.01.

For the cell-type proximity mIF analyses, we used a Wilcoxon rank-sum test to compare proximity calculations between older and younger cohorts, followed by Benjamini–Hochberg correction.

## Reporting summary

Further information on research design is available in the Nature Portfolio Reporting Summary linked to this article.

## Data availability

All data necessary to interpret and verify the analyses in this study are available publicly through the original publication or as Source Data. The publicly available gene expression data for TNBC and ER⁺ breast cancer for METABRIC and Basal and Luminal A breast cancers in the TCGA were accessed through the cBioPortal[80–82]. The TNBC and ER⁺ single-cell RNA sequencing data used in this study are publicly available and were accessed through the GEO, under accession no. GSE176078. Source data for the figures have been provided in Excel format, citing related figures in the file. The larger gene expression files for METABRIC used in the analysis for Fig. 1 can be found at https://doi.org/10.6084/m9.figshare.27242253.v1 (ref. 83) and https://doi.org/10.6084/m9.figshare.27242256.v1 (ref. 84). The remaining data supporting the findings of this study, including the mIF proximity data, are part of a larger unpublished clinical cohort. These data are available upon reasonable request.

## Code availability

All analysis scripts are available at https://github.com/adrienneparsons/BC_singlecell_age. We also include a pseudocode document describing ASPEN in plain language in the same GitHub repository. The folder 'Reference Scripts' includes a more basic R script of the ASPEN framework to facilitate implementation for other scRNA-seq datasets.

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

## Acknowledgements

A.P. is supported by a National Institutes of Health grant no. T15LM007092 (principal investigator: N. Gehlenborg). M.S. is supported by a Breast Cancer Research Fellowship from the American Association for Cancer Research (AACR) (23-40-12 SPAS). A.S., J.C. and H.Z. are supported by Deborah and John McMurtrie, the Petre Foundation and grants from the Breast Cancer Research Foundation (BCRF-23-209), the National Health and Medical Research Council (NHMRC) (APP2018440) and National Breast Cancer Foundation (IIRS-23-074). P.v.G. is supported by the National Cancer Institute Innovative Molecular Analysis Technologies Program (R33 CA278393), the Edward P. Evans Foundation, the Vera and Joseph Dresner Foundation, the MPN Research Foundation, an American Cancer Society Research Scholar Grant (no. RSG-24-1318769-01-CDP, https://doi.org/10.53354/ACS.RSG-24-1318769-01-CDP.pc.gr.222076), the Hevolution/American Federation for Aging Research New Investigator Award in Aging Biology and Geroscience Research, the Krantz Family Center for Cancer Research and the Ludwig Center at Harvard. S.S.M. is supported by the Samuel Waxman Cancer Research Foundation in partnership with the Mark Foundation for Cancer Research, Victoria's Secret Global Fund for Women's Cancer Rising Innovator Research Grant, in Partnership with Pelotonia and the AACR (23-30-73-MCAL), a DOD/CDMRP/BCRP Era of Hope Scholar Expansion Award (W81XWH-14-1-0191) and the National Cancer Institute (R01 CA279959). We thank members of the Mittendorf, Van Galen and McAllister labs for helpful discussion. We are grateful to the patients who participated in the clinical trials and donated tissue.

## Author contributions

A.P., P.v.G. and S.S.M. designed the study. A.P. designed the methodology. A.P. and E.S.C. performed the computational analysis. M.M. performed the experimental mIF staining and analysis. H.Z. and J.C. performed the TMA spatial analysis. M.S., B.K., B.B.K., A.S., R.A.F., E.A.M., P.v.G. and S.S.M. provided input. P.v.G. and S.S.M. led the study. A.S. led the TMA spatial analysis. A.P., P.v.G. and S.S.M. wrote the paper. P.v.G. and S.S.M. acquired the funding. All authors edited the paper.

## Competing interests

The authors declare no competing interests.

## Additional information

**Extended data** is available for this paper at https://doi.org/10.1038/s43587-025-00984-1.

**Correspondence and requests for materials** should be addressed to Peter van Galen or Sandra S. McAllister.

[1]Division of Hematology, Department of Medicine, Brigham and Women's Hospital, Boston, MA, USA. [2]Department of Medicine, Harvard Medical School, Boston, MA, USA. [3]Department of Medical Oncology, Dana-Farber Cancer Institute, Boston, MA, USA. [4]Breast Oncology Program, Dana-Farber Brigham Cancer Center, Boston, MA, USA. [5]Oncological Pathology and Bioinformatics Research Group, Hospital Verge de la Cinta, Institut d'Investigació Sanitària Pere Virgili, Universitat Rovira i Virgili, Tortosa, Spain. [6]Cancer Ecosystems Program, Garvan Institute of Medical Research, Darlinghurst, New South Wales, Australia. [7]School of Clinical Medicine, Faculty of Medicine and Health, University of New South Wales, Sydney, New South Wales, Australia. [8]Department of Pathology, Mass General Brigham, Boston, MA, USA. [9]Division of Breast Surgery, Department of Surgery, Brigham and Women's Hospital, Boston, MA, USA. [10]Breast Cancer Program, Dana-Farber/Harvard Cancer Center, Boston, MA, USA. [11]Broad Institute of Harvard and MIT, Cambridge, MA, USA. [12]Harvard Stem Cell Institute, Cambridge, MA, USA. [13]Ludwig Center at Harvard, Harvard Medical School, Boston, MA, USA. [14]These authors jointly supervised this work: Peter van Galen, Sandra S. McAllister. ✉e-mail: pvangalen@bhw.harvard.edu; smcallister1@bwh.harvard.edu

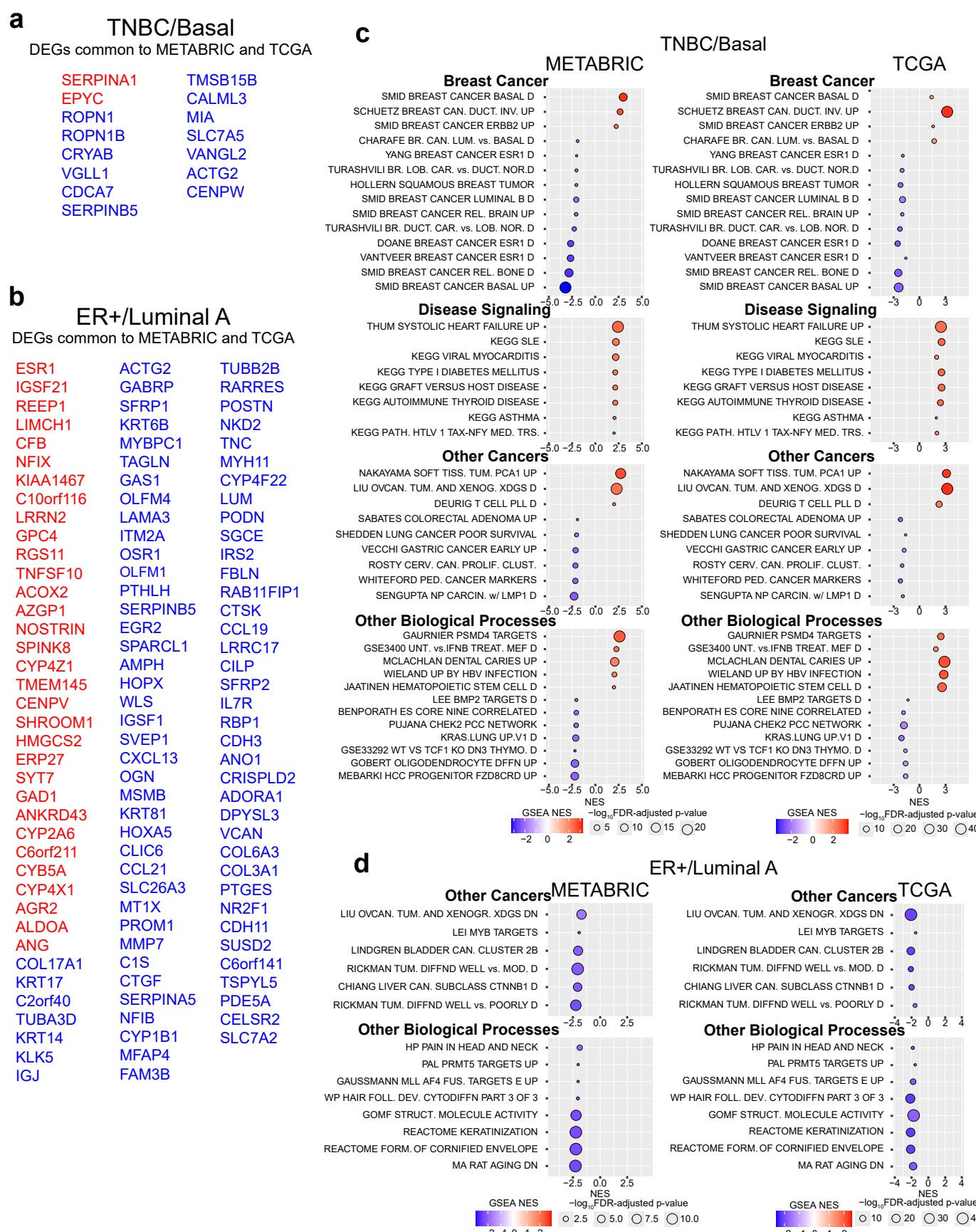

**Extended Data Fig. 1 | See next page for caption.**

**Extended Data Fig. 1 | Additional age-related functional gene set enrichments in TNBC and ER+ breast cancer (related to Fig. 1). a, b,** Differentially expressed genes that were identified as common between TNBC (METABRIC) and basal (TCGA) tumors (**a**) and ER+ (METABRIC) and Luminal A (TCGA) tumors (**b**). Red text indicates common enrichment in the >65 age group, blue text indicates enrichment in the <45 age group. Statistical significance was determined using an empirical Bayes-moderated two-sided t test. **c, d,** GSEA dot plots show pathways that were significantly enriched in <45 (blue) or > 65 (red) age cohorts in TNBC/Basal (**a**) and ER + /Luminal A breast cancer (**b**) from METABRIC/TCGA.). Statistical significance and normalized enrichment is determined using a permutation-based null distribution, per the calculations of the *fgsea* R package. Dot size is proportional to the -log$_{10}$Benjamini-Hochberg-adjusted p-value; color intensity represents magnitude of normalized enrichment score (NES), according to indicated scales. Pathways were manually grouped by functional similarity.

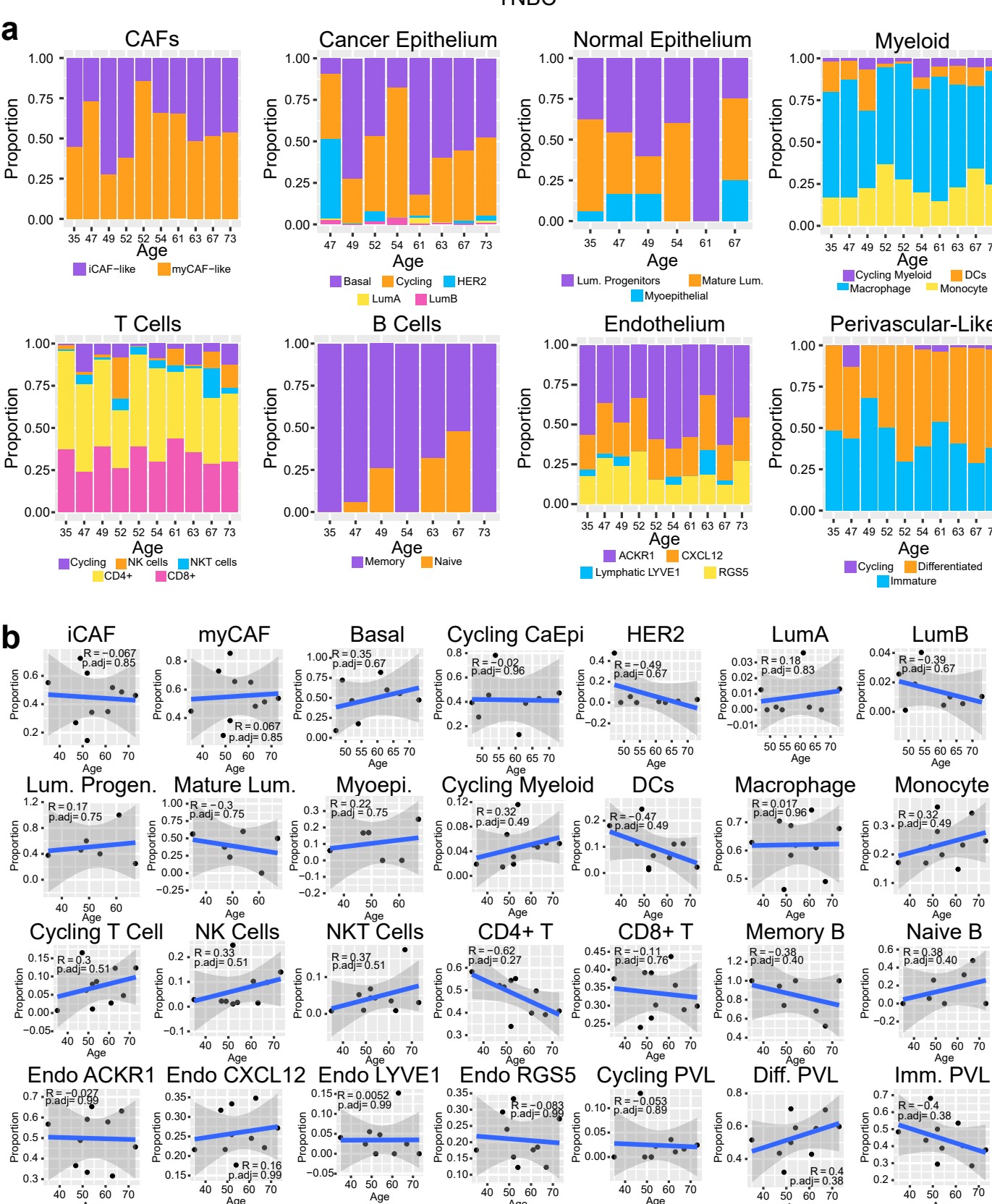

**Extended Data Fig. 2 | Changes in abundance of epithelial, stromal, and immune cell populations with age in TNBC. a**, Stacked bar charts representing the indicated minor cell populations as a proportion of respective major cell populations in TNBC tumors from the human breast cancer atlas[23]. Samples are ordered along x axes by donor age, from youngest to oldest; samples that did not have any cells of a given major cell type are excluded from the plots. **b**, Scatter plots representing Pearson correlation between donor age and cell proportion (from **a**) for each of 28 cell subpopulations. Plots depict the line of best fit and 95% confidence interval following Pearson correlation analysis. p.adj indicates Benjamini-Hochberg-adjusted two-sided Pearson's correlation coefficients.

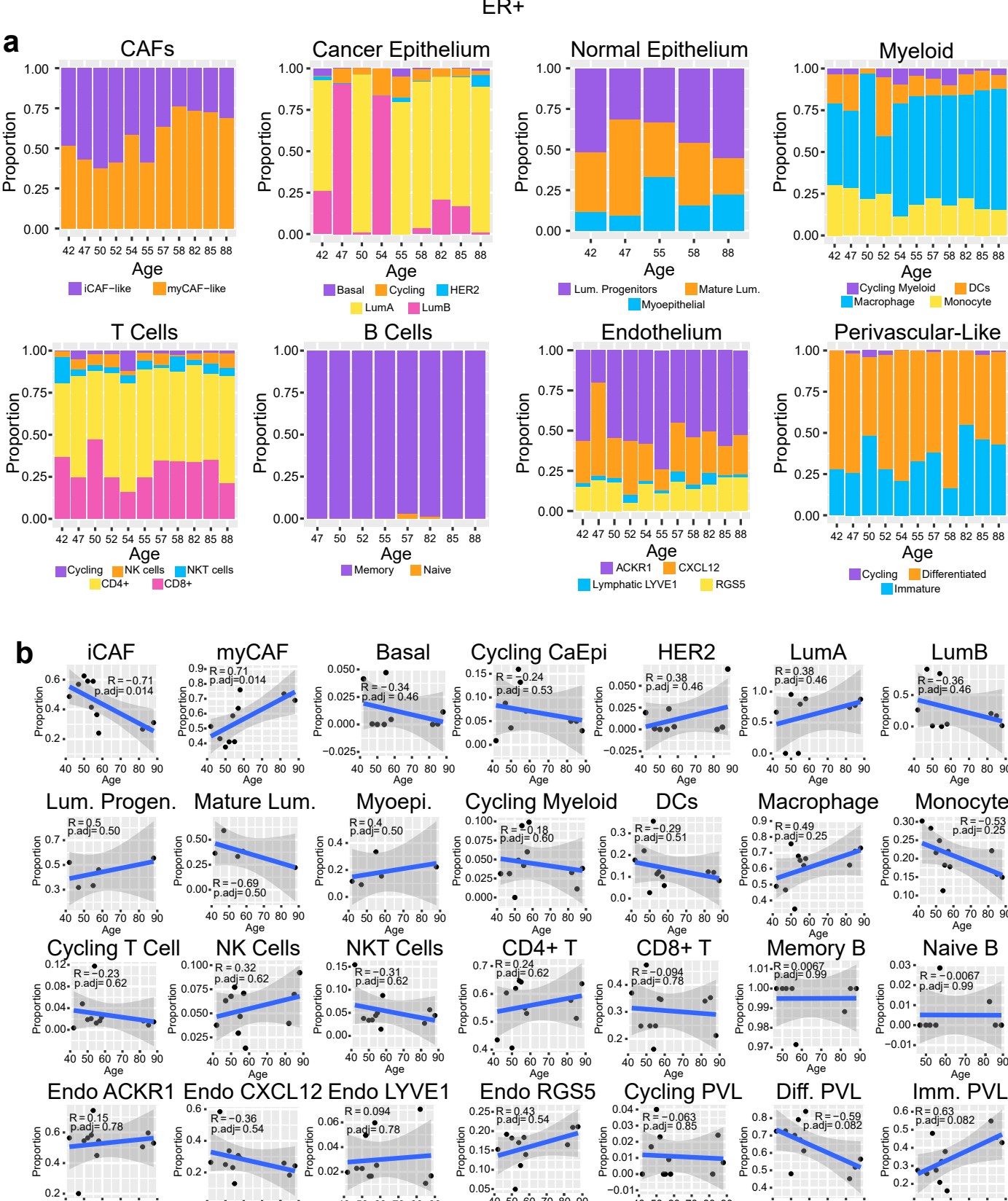

**Extended Data Fig. 3 | Changes in abundance of epithelial, stromal, and immune cell populations with age in ER+ breast cancer. a**, Stacked bar charts representing the indicated minor cell populations as a proportion of respective major cell populations in ER+ breast tumors from the human breast cancer atlas[23]. Samples are ordered along x axes by donor age, from youngest to oldest; samples that did not have any cells of a given major cell type are excluded from the plots. **b**, Scatter plots representing Pearson correlation between donor age and cell proportion (from **a**) for each of 28 cell subpopulations Plots depict the line of best fit and 95% confidence interval following Pearson correlation analysis. p.adj indicates Benjamini-Hochberg-adjusted two-sided Pearson's correlation coefficients.

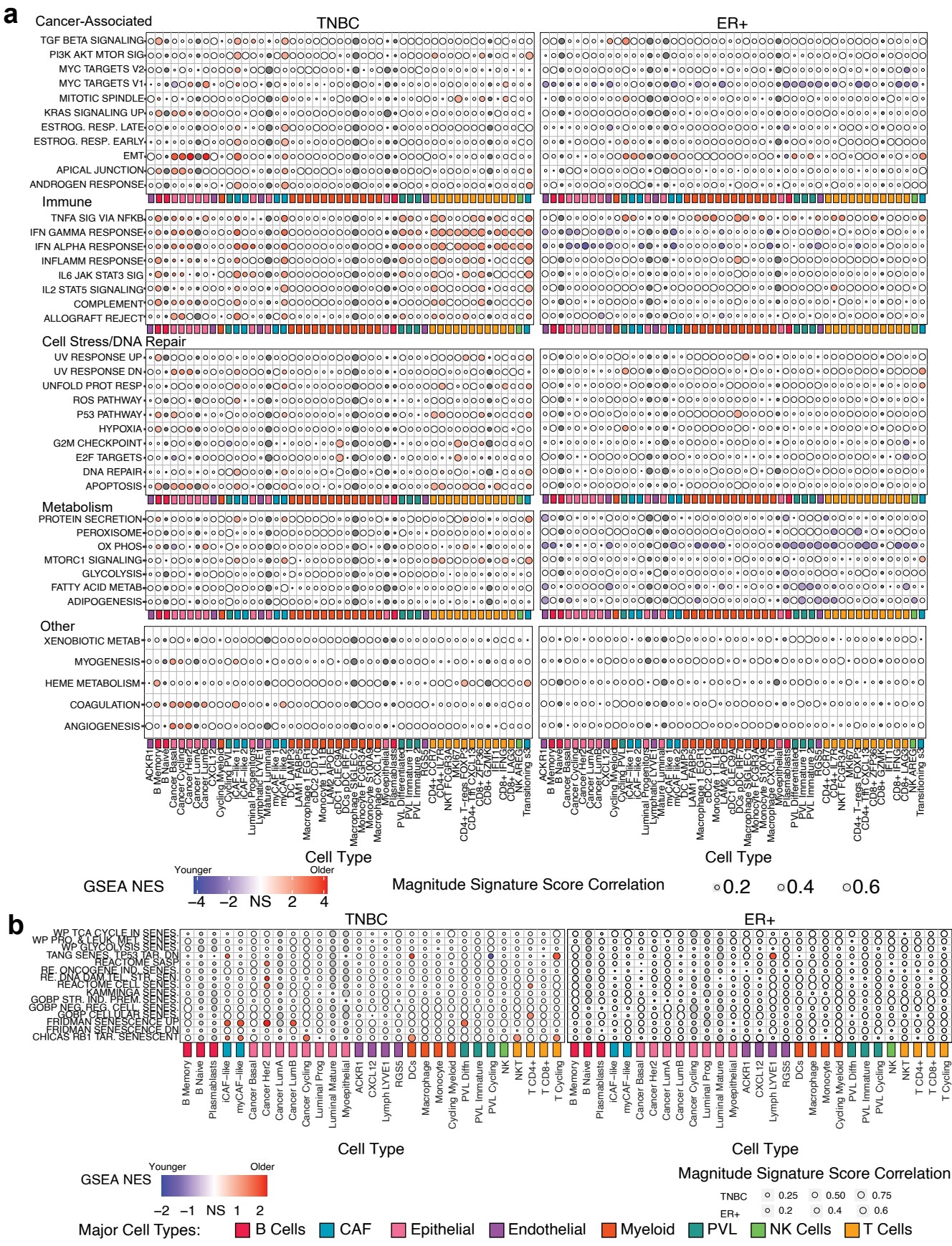

**Extended Data Fig. 4 | See next page for caption.**

**Extended Data Fig. 4 | ASPEN analysis of 49 cell subsets and senescence-related gene signatures (related to Fig. 3). a**, Results from ASPEN analysis of the 49 cell subsets from the breast cancer scRNA-seq atlas dataset and Hallmark gene sets (Human MSigDB), yielding cell-specific age-related programs (ARPs) in TNBC (left) ER+ breast cancer (right). Cell type subsets (color coded by major cell type groups) are represented on x-axes and Hallmark pathways with at least one statistically significant cell type are on the y-axes. ARPs were manually grouped into biologically similar processes. **b**, Results from ASPEN for 29 cell types (x-axis) and senescence-associated pathways from the MSigDB curated (C2) and ontology (C5) gene sets (y-axis) in TNBC (left) and ER+ (right) breast cancer. Cell types are color coded by indicated major cell type groups. **a**,**b**, Bubble color indicates normalized enrichment score (NES) of age-associated GSEA analysis, with color intensity indicating magnitude of enrichment according to the indicated scale. Statistical significance of NES is determined at a threshold of adjusted p-value < 0.05, whereby significance must be achieved in both *fgsea*-derived permutation test and *gage*-derived two-sided Welch's t-test–style parametric gene set test (see Methods). Red indicates significant enrichment in older donors; blue indicates significant enrichment in younger donors; white indicates not significant (NS); gray indicates cell types were present in <50% of donors and were excluded from analysis. Circle size indicates magnitude of Seurat-identified enrichment score correlation to age according to the scale. TNF = tumor necrosis factor, SIG = signaling, IFN = Interferon, RESP = response, SIGNAL = Signaling, REJECTN = rejection, OX PHOS = oxidative phosphorylation, METAB = metabolism, TGF = transforming growth factor, ESTRGN = estrogen, EMT = epithelial to mesenchymal transition, DN = down, UNFOLD PROT RESP = Unfolded Protein Response, CAF = cancer associated fibroblast, PVL = perivascular-like cells.

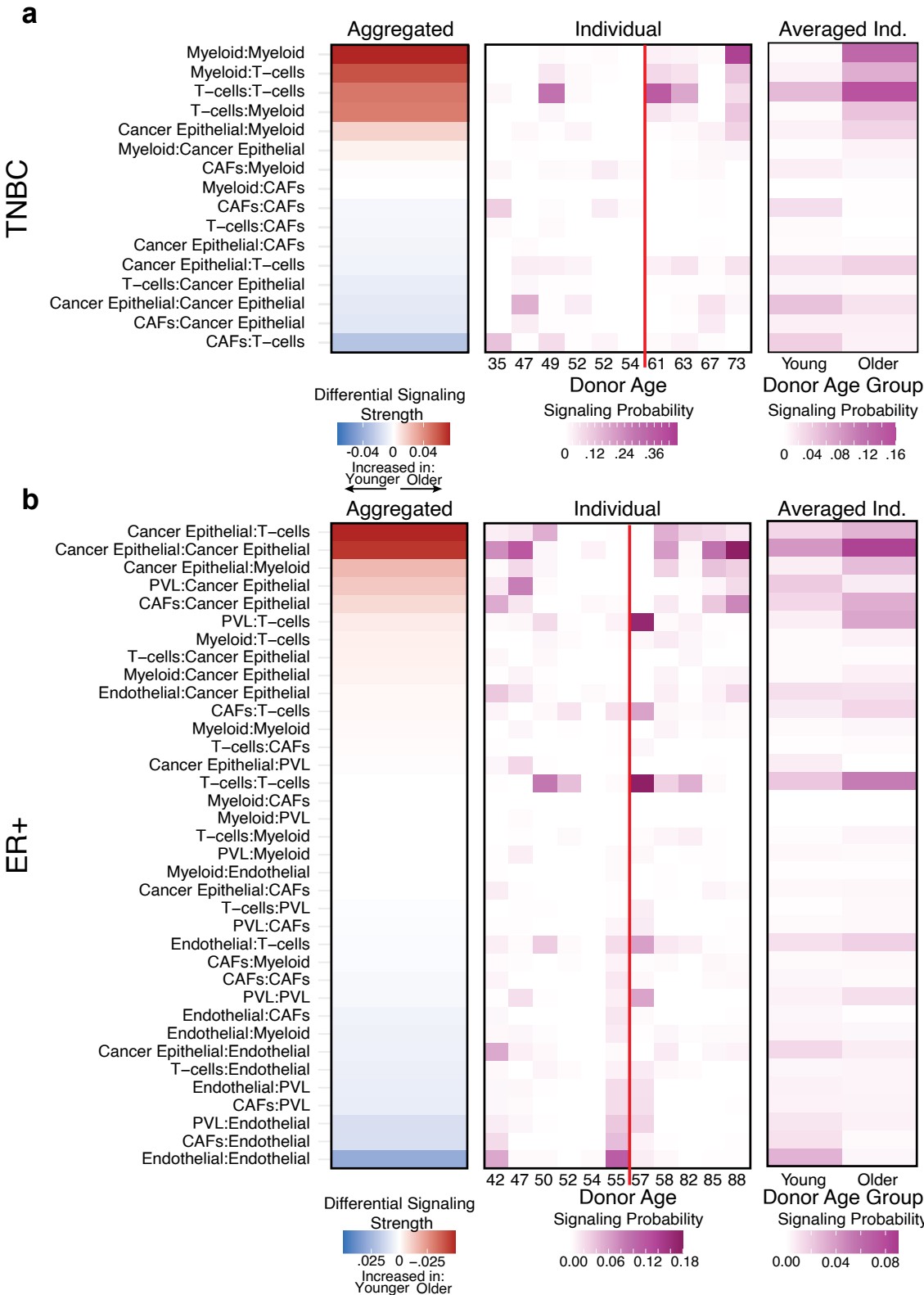

**Extended Data Fig. 5 | Comparison of CellChat from individuals to aggregate cohorts (related to Fig. 5). a, b,** Heat maps show differential signaling probabilities between major cell groups of interest in TNBC (**a**) and ER+ breast cancer (**b**) after either aggregating single-cell RNA seq data from donors ≤55 years and >55 years (left), or calculating signaling probabilities for each donor individually, ordered from youngest to oldest (middle), or after averaging the individual signaling probabilities for donors ≤55 years and >55 years (right). Red color indicates higher signaling strength in the >55 age group, blue color represents higher signaling strength in the ≤55 age group. Deeper color indicates stronger signaling probability.

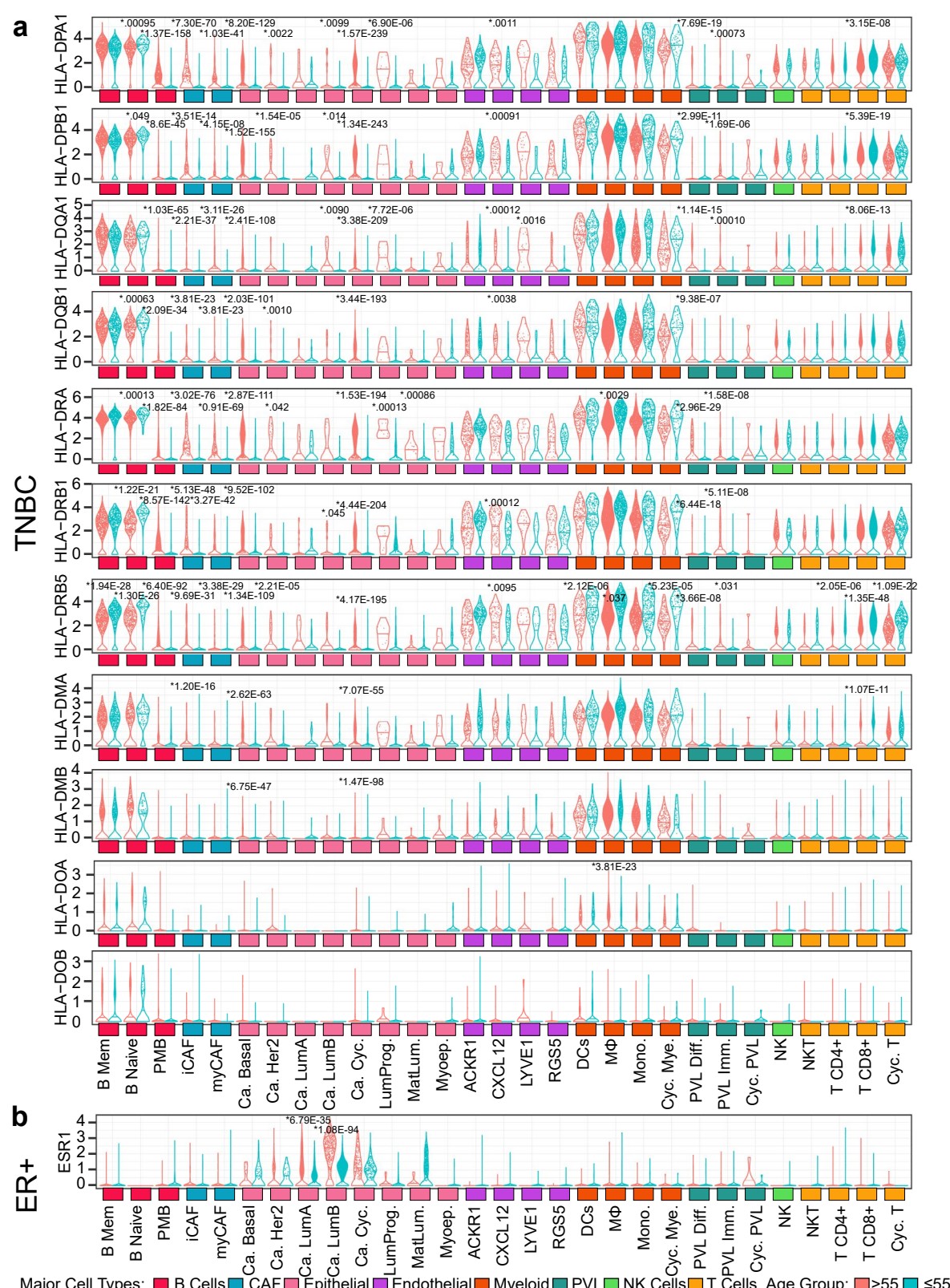

**Extended Data Fig. 6 | Cell type-specific age-related expression levels of MHC class II and ESR1 genes. a, b**, Violin plots show age-stratified expression of MHC II genes in TNBC (**a**) and ESR1 in ER+ breast cancer (**b**). Log-normalized gene expression levels (y axes) are indicated for each cell type (x axes). Red violins represent data from donors >55 years; blue represents data from donors ≤55 years. Adjusted two-sided Wilcoxon Rank Sum test between ≤55 and >55 for a given cell type and gene is indicated when below significance threshold of 0.05.

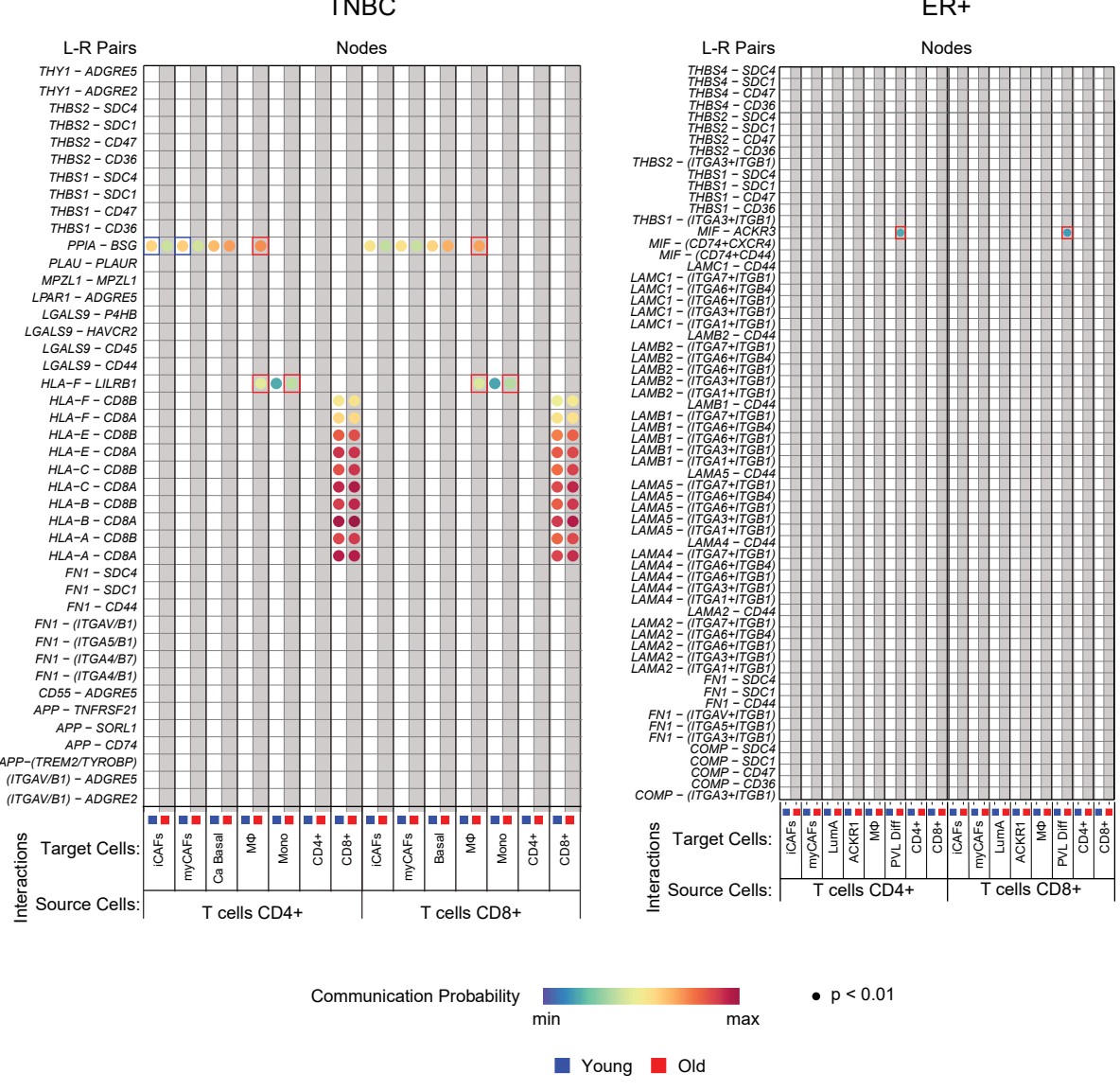

**Extended Data Fig. 7 | Age-associated signaling networks specific to CD4+ and CD8 + T cells in TNBC ER+ breast cancer (related to Fig. 6).** Bubble plots representing the communication probability for each indicated ligand-receptor pair between CD4+ or CD8 + T cells and indicated target cells in TNBC (left) and ER+ breast cancer (right). Rows depict the ligand-receptor pairs and signaling pathways; columns depict specific source-target cell interactions in the ≤55 cohort (blue) or >55 cohort (red). Communication probabilities are represented by a color scale, with minimum values colored deep blue, increasing values depicted as green, then yellow, then orange, and maximum values as deep red. Each bubble represents a signaling node predicted to be active with permutation test-based p value < 0.01 according to CellChat probability calculations[30]. Colored boxes around bubbles indicate signaling nodes that were differentially enriched by at least 1.2- fold in either the younger (blue boxes) or older (red boxes) cohort.

# Reporting Summary

## Statistics

For all statistical analyses, confirm that the following items are present in the figure legend, table legend, main text, or Methods section.

| n/a | Confirmed | |
|---|---|---|
| ☐ | ☒ | The exact sample size (*n*) for each experimental group/condition, given as a discrete number and unit of measurement |
| ☐ | ☒ | A statement on whether measurements were taken from distinct samples or whether the same sample was measured repeatedly |
| ☐ | ☒ | The statistical test(s) used AND whether they are one- or two-sided<br>*Only common tests should be described solely by name; describe more complex techniques in the Methods section.* |
| ☐ | ☒ | A description of all covariates tested |
| ☐ | ☒ | A description of any assumptions or corrections, such as tests of normality and adjustment for multiple comparisons |
| ☐ | ☒ | A full description of the statistical parameters including central tendency (e.g. means) or other basic estimates (e.g. regression coefficient) AND variation (e.g. standard deviation) or associated estimates of uncertainty (e.g. confidence intervals) |
| ☐ | ☒ | For null hypothesis testing, the test statistic (e.g. *F*, *t*, *r*) with confidence intervals, effect sizes, degrees of freedom and *P* value noted<br>*Give P values as exact values whenever suitable.* |
| ☒ | ☐ | For Bayesian analysis, information on the choice of priors and Markov chain Monte Carlo settings |
| ☒ | ☐ | For hierarchical and complex designs, identification of the appropriate level for tests and full reporting of outcomes |
| ☐ | ☒ | Estimates of effect sizes (e.g. Cohen's *d*, Pearson's *r*), indicating how they were calculated |

*Our web collection on statistics for biologists contains articles on many of the points above.*

## Software and code

Policy information about availability of computer code

| Data collection | Image analysis was performed in Fiji/ImageJ v2.9.0 and QuPath v0.2.3. |
|---|---|
| Data analysis | All data analysis was conducted in R v4.4.1. Seurat v5 was used for CellChat analyses; all other Seurat analysis was completed in v4.4.0. CellChat v2.1.2. gage v2.54.0. fgsea v1.30.0. limma v3.60.6. ggplot2v 3.5.1 msigdbr v25.1.1 All analysis scripts are available at https://github.com/adrienneparsons/BC_singlecell_age. We also include a pseudocode document describing ASPEN in plain language in the same GitHub Repository. The folder "Reference Scripts" includes a more basic R script of the ASPEN framework to facilitate implementation for other single-cell RNA-seq datasets. |

For manuscripts utilizing custom algorithms or software that are central to the research but not yet described in published literature, software must be made available to editors and reviewers. We strongly encourage code deposition in a community repository (e.g. GitHub). See the Nature Portfolio guidelines for submitting code & software for further information.

## Data

Policy information about availability of data

All manuscripts must include a data availability statement. This statement should provide the following information, where applicable:

- Accession codes, unique identifiers, or web links for publicly available datasets
- A description of any restrictions on data availability
- For clinical datasets or third party data, please ensure that the statement adheres to our policy

All data necessary to interpret and verify the analyses in this study are available publicly through the original publication or as Source Data. The publicly available gene expression data for TNBC and ER+ breast cancer for METABRIC and Basal and Luminal A breast cancers in TCGA were accessed through cBioPortal 81–83. The TNBC and ER+ single-cell RNA sequencing data used in this study are publicly available and were accessed through GEO Accession number GSE176078 (https://www.ncbi.nlm.nih.gov/geo/query/acc.cgi?acc=GSE176078). Source data for figures have been provided in Excel format, citing related Figures in the file. The larger, gene expression files for METABRIC used in the analysis for Figure 1 can be found at https://figshare.com/articles/dataset/2024_METABRIC_TNBC_csv/27242253?file=4983452 and https://figshare.com/articles/dataset/2024_METABRIC_ER_csv/27242256?file=49834524

Remaining data supporting the findings of this study, including the mIF proximity data are part of a larger, unpublished clinical cohort. These data are available on reasonable request. All patients included provided written informed consent.

## Research involving human participants, their data, or biological material

Policy information about studies with human participants or human data. See also policy information about sex, gender (identity/presentation), and sexual orientation and race, ethnicity and racism.

| | |
|---|---|
| Reporting on sex and gender | Analyses were limited to females with breast cancer. Gender information was not available. |
| Reporting on race, ethnicity, or other socially relevant groupings | Not applicable |
| Population characteristics | Breast cancer patient age information is described in Source Data Tables 1, 3, and 4. |
| Recruitment | Not Applicable. |
| Ethics oversight | Tissue for the older cohorts (>70 years, grade I-III disease) were provided for secondary use in a de-identified manner under Mass General Brigham institutional review board approval 2021P001031, from the ELEVATE (clinical trial registration number: NCT03818087) and ADVANCE (clinical trial registration number: NCT03858322) studies. Original tissue collection was performed with institutional review board approval from all participating institutions and following the Declaration of Helsinki. All patients provided written informed consent. Tissue for the younger cohorts (<45 years, grade I-III) was commercially available from AMSBio. The ER+ TMA samples were from the Breast Boost cohort recruited to the St George Breast Boost study between 1998 and 2003 (Clinical Trials Registry NCT00138814). The TNBC TMA cohort consists of TNBC cases diagnosed between 2004 and 2019 at St George Hospital, Sydney, Australia. The TNBC cohort was not collected under a clinical trial. Ethics approval was granted by the South Eastern Sydney Local Health District Human Research Ethics Committee at the Prince of Wales Hospital, Sydney (Boost: HREC 96/16 and TNBC: HREC 2018/ETH00138) who granted a waiver of consent to perform research analyses on the tissue blocks. All methods were performed in accordance with the relevant institutional guidelines and regulations. |

Note that full information on the approval of the study protocol must also be provided in the manuscript.

# Field-specific reporting

Please select the one below that is the best fit for your research. If you are not sure, read the appropriate sections before making your selection.

☒ Life sciences        ☐ Behavioural & social sciences        ☐ Ecological, evolutionary & environmental sciences

For a reference copy of the document with all sections, see nature.com/documents/nr-reporting-summary-flat.pdf

# Life sciences study design

All studies must disclose on these points even when the disclosure is negative.

| | |
|---|---|
| Sample size | Sample sizes for computational methods relied on availability of publicly available datasets used in the analysis. Sample sizes for experimental validation relied on patient enrollment and sample availability. No statistical methods were used to determine sample sizes. The METABRIC cohorts consisted of n = 63 TNBC >65, n = 50 TNBC <45, n = 386 ER+ >65, and n = 86 ER+ <45. The TCGA cohort consisted of n = 37 basal >65, n = 30 basal < 45, n = 152 Luminal A >65, n = 68 Luminal A < 45. Cell Composition analyses, ASPEN, and CellChat analyses were performed on a cohort of n = 10 TNBC and n = 11 ER+ patients. Tissue mIF cohorts include n = 6 TNBC > 70, n = 5 TNBC < 45, n = 7 ER+ > 70, and n = 5 ER+ < 45. TMA analysis cohorts included n = 127 TNBC >55, n = 94 TNBC <=55, n = 237 ER+ > 55, n = 264 ER+ <=55. |
| Data exclusions | For computational analyses of the single-cell RNA seq atlas, we excluded HER2+ samples (n=5), because there were insufficient sample numbers and age ranges for ASPEN and downstream analyses. For bulk transcriptomic analyses of METABRIC and TCGA, we therefore also |

excluded HER2+ patients, as well as patients with late stage or stage 0 disease. We also established prior to analysis to exclude METABRIC and TCGA patients between ages 45 and 65 years to align with established clinical risk. For TMAs, to account for potential misclassified cells, we excluded myCAFs, iCAFs, and endothelial cells identified within tumor regions, as well as tumor cells detected within stromal regions from the spatial analysis. TMA cores with <10% or >90% stromal region were excluded due to insufficient data, often associated with artifacts or missing morphology. In some ASPEN and CellChat analyses, exclusion criteria were pre-established, and some cell types were not analyzed due to insufficient representation of the cell type across the whole cohort, as described in the text.

| Replication | All experiments and analyses were performed with biological replicates. Sample sizes are outlined in Source Data and within the text. All available data was analyzed, barring exclusions as described above. For mIF analyses, 5-9 images were analyzed per tumor. For TMA mIF, . each core was imaged once; mean cores analyzed for the ≤55 ER+ cohort: 2.5 (1-3 per patient); mean cores analyzed for >55 ER+ cohort: 2.27 (1-3 per patient); mean cores analyzed for the ≤55 TNBC cohort: 2.28 (1-3 per patient); mean cores analyzed for >55 TNBC cohort: 2.13 (1-3 per patient). |

| Randomization | Samples were analyzed based on established disease subtypes and age ranges, and all available tissue was analyzed per cohort, barring exclusions as described above. Therefore, randomization was not necessary for this study. |

| Blinding | Tissue-based assays and analyses were performed in a blinded fashion. For computational analysis of existing data, blinding was not performed, as it was necessary to classify samples into known subtype and age categories. |

# Reporting for specific materials, systems and methods

We require information from authors about some types of materials, experimental systems and methods used in many studies. Here, indicate whether each material, system or method listed is relevant to your study. If you are not sure if a list item applies to your research, read the appropriate section before selecting a response.

## Materials & experimental systems

| n/a | Involved in the study |
|---|---|
| ☐ | ☒ Antibodies |
| ☒ | ☐ Eukaryotic cell lines |
| ☒ | ☐ Palaeontology and archaeology |
| ☒ | ☐ Animals and other organisms |
| ☒ | ☐ Clinical data |
| ☒ | ☐ Dual use research of concern |
| ☒ | ☐ Plants |

## Methods

| n/a | Involved in the study |
|---|---|
| ☒ | ☐ ChIP-seq |
| ☒ | ☐ Flow cytometry |
| ☒ | ☐ MRI-based neuroimaging |

## Antibodies

| Antibodies used | For TMAs: panCK (1:2000, clone: AE1/AE3, host: Mouse, ab27988; Abcam), PDGFRβ (1:1000, CD140b, clone: Y92, host: Rabbit, ab32570; Abcam), αSMA (1:500, polyclonal, host: Rabbit, ab5694; Abcam), CD146 (1:1250, clone: EPR3208, host: Rabbit, ab75769; Abcam), THY1 (1:4000, CD90, clone: EPR3133, host: Rabbit, ab133350; Abcam), CD8 (1:1000, clone: C8/144B, host: Mouse, MA5-13473; Invitrogen), PD-1 (1:50, clone: EPR4877(2), host: Rabbit, ab137132; Abcam), and CD31 (1:100, clone: JC70A, host: Mouse, M0823; Agillent Technologies/DAKO). Lot numbers are unavailable; For additional mIF staining: Pan cytokeratin (1:100, clone: AE1/AE3, host: mouse, NBP2-2949; Lot 371P240806; Novus Biologicals), Vimentin (1:50, clone: SP20, host: rabbit, MA5-16409; Lot ZH4438378; Invitrogen), CD31 (1:50, clone: RM247, host: rabbit, MA5-33063; Lot ZL4575373; Invitrogen), COX4 (1:50, clone: 4D11-B3-E8, host: mouse, 11967S; Lot 4; Cell Signaling Technologies). Secondary antibodies: Donkey anti-Rabbit IgG (H+L) Highly Cross-Adsorbed Secondary Antibody, Alexa Fluor 488 (Invitrogen, A-21206, Lot 2330673), Goat anti-Mouse IgG (H+L) Highly Cross-Adsorbed Secondary Antibody, Alexa Fluor 594 (Invitrogen, A-21145, Lot 1736995), Goat anti-Mouse IgG (H+L) Highly Cross-Adsorbed Secondary Antibody, Alexa Fluor 647 (Invitrogen, A-21240, Lot 1772672), Goat anti-Rabbit IgG (H+L) Highly Cross-Adsorbed Secondary Antibody, Alexa Fluor 647 (Invitrogen, A-21244, Lot 2086678). |

| Validation | All primary antibodies were validated according to the manufacturer's associated product datasheet. For TMAs, Immunohistochemical (IHC) staining was used as a control to compare with mIF staining. Normal tissue cores were included as internal controls to ensure consistent staining intensity across all slides. During the multiplex optimization phase, antibody concentrations were further refined to standardize signal intensity where needed. |

## Plants

Seed stocks

Not applicable

Novel plant genotypes

Not applicable

Authentication

Not applicable

