## [Peer Review File · Nature Aging]

Cell Populations in Human Breast Cancers are Molecularly and Biologically Distinct with Age

Corresponding Author: Dr Sandra McAllister

This manuscript has been previously reviewed at another journal that is not operating a transparent peer review scheme. The manuscript was considered suitable for publication without further review at Nature Aging.

Version 0:

Reviewer comments:

Reviewer #1

(Remarks to the Author)

Overall, the manuscript is greatly improved and the robustness of the conclusions increased with additional datasets. There are still a few more concerns that need to be addressed.

Reviewer comment 1: The increase in FOXA1 in older TNBC patients (Figure 1a), as this is typically expressed in luminal cells and associated the androgen signaling. The genes enriched in younger TNBC also appear to be more basal (KRT6B, SOX10, ELF5, FOXC1). Is there any other evidence of luminal differentiation in older TNBC?

Response 1: We specifically interrogated the TNBC cancer epithelial cells in the scRNAseq breast cancer atlas for various genes associated with luminal and basal differentiation status in breast epithelium, as well as Hallmark and MSigDB pathways related to breast differentiation. The following table summarizes our results: These results reveal a compelling trend toward loss of luminal and acquisition of basal phenotypes with age in the TNBC cancer epithelium. These data are consistent with studies that evaluated luminal/basal phenotypes in normal breast epithelium with age, particularly (6,7), showing loss of epithelial lineage fidelity and with markers in normal luminal epithelial cells with age, which presage cancer susceptibility, upregulation of basal establishing AXL as a regulator of epithelial cell plasticity in the normal mammary gland as well as driving stemness in basal breast cancer.

Although, the METABRIC analyses show enrichment of KRT6B, FOXC1, SOX10, and ELF5 in younger and enrichment of FOXA1 in older TNBC patients, none of these genes was significantly differentially expressed in the TCGA dataset. We believe this inconsistency between datasets could reflect the limitations of bulk transcriptional profiling/sequencing, as it is not clear which cells differentially express these factors in the METABRIC and TCGA datasets. For example, examination of the scRNAseq breast cancer atlas data indicates that the normal myoepithelium contributes the greatest expression of KRT14 and KRT6B signals in the young cohorts.

Reviewer Response 1: In response to this, the authors evaluated luminal and basal differentiation in TNBC from the scRNAseq breast cancer atlas relative to age. They present a table that shows decreased luminal differentiation with age with a loss of FOXA1. These results are now opposite of the METABRIC findings in figure 1. The lack of significance in the TCGA data may be due to small sample size (30 vs. 37). We agree with the reviewer that different cell types contribute different markers, however the bulk of the expression in bulk tumors should be from the tumor cells themselves, expressing either luminal or basal markers. Perhaps analysis of a larger dataset like the Fudan TNBC dataset will provide further evidence for enrichment of loss of luminal markers with age (see following comment).

Reviewer comment 2: The conclusions are valid however the robustness of the analysis is limited to few scRNA samples, particularly when stratifying in you young and old for the cell-cell interactions. Therobustness of the age associated transcript changes identified in METABRIC could be increasedwith validation in the TCGA (for TNBC and ER+) and the Fudan TNBC cohort (PMID: 30853353.)

Response 2: As suggested, we also evaluated the feasibility of incorporating the Fudan TNBC cohort into our analysis. However, the available expression data from this cohort is only provided as raw FASTQ files in the Sequence Read Archive,

rather than as processed, normalized count matrices. Integrating raw FASTQ data would require reprocessing with potential batch effects and methodological discrepancies compared to the METABRIC and TCGA datasets. Given these limitations, and the high concordance we observed between larger, multi-institutional METABRIC and TCGA, we prioritized these two robust, complementary datasets for ensuring broader relevance of our initial findings.

Reviewer Response 2: We agree with the authors that analyzing fastq would be too cumbersome for the Fudan dataset. However, the authors have provided processed data on NODE. Below is the data availability statement from the manuscript. The inclusion of the Fudan cohort may provide more support for the confounding data with the shift in the luminal subtype with age.

The accession number for all the data reported in this paper is NODE: OEP000155. All data can be viewed in The National Omics Data Encyclopedia (NODE) (<http://www.biosino.org/node>) by pasting the accession (OEP000155) into the text search box or through the URL: <http://www.biosino.org/node/project/detail/OEP000155>

(Remarks on code availability)

Reviewer #2

(Remarks to the Author)

The authors have thoroughly addressed my previous concerns. They have expanded the dataset, clarified methodological details, and provided additional analyses and explanations that strengthen the conclusions. I appreciate the thoughtful revisions, which improve the clarity, rigor, and overall impact of the study.

(Remarks on code availability)

The code is well prepared, clearly documented, and facilitates reproducibility.

Reviewer #3

(Remarks to the Author)

The authors have made substantial and commendable revisions in response to the initial round of reviews. Notably, they have increased the sample size and incorporated two independent clinical cohorts to validate key findings derived from the Aspen algorithm. Furthermore, they have strengthened their conclusions by employing multiplex fluorescence imaging on tumor tissue from an additional independent cohort, which includes both older and younger women with triple-negative breast cancer (TNBC) or ER-positive disease. These additions significantly enhance the robustness and relevance of the study. Overall, I am impressed by the depth and rigor of the revisions.

However, I would like to draw attention to the discussion section, specifically lines 646–660. The authors still acknowledge the limitations of their sample size, though they do not adequately emphasize the fact that they have addressed this concern through validation in orthogonal cohorts. This is a critical strength of the revised manuscript and should be highlighted more explicitly. Doing so would not only reinforce the credibility of their findings but also provide a valuable roadmap for other researchers facing similar challenges in validating gene-level predictions in limited-sample contexts.

(Remarks on code availability)

Version 1:

Reviewer comments:

Reviewer #1

(Remarks to the Author)

The authors have thoroughly addressed my previous concerns.

(Remarks on code availability)

We value the reviewers' time and thoughtful consideration of the revised manuscript. We are pleased to know there is consensus that our manuscript is significantly improved. We did our best to address the few remaining points and provide our responses in blue text below.

Reviewers' Comments:

Reviewer #1 (Remarks to the Author):

Bold: reviewer's current comments upon re-review

Purple: original reviewer comment in round 1 of reviews

Grey: original author response in round 1 of reviews

Blue: authors' current response to re-review

Overall, the manuscript is greatly improved and the robustness of the conclusions increased with additional datasets. There are still a few more concerns that need to be addressed.

Reviewer 1 original comment 1: The increase in FOXA1 in older TNBC patients (Figure 1a), as this is typically expressed in luminal cells and associated the androgen signaling. The genes enriched in younger TNBC also appear to be more basal (KRT6B, SOX10, ELF5, FOXC1). Is there any other evidence of luminal differentiation in older TNBC?

Author original response 1 (with corrections noted): We specifically interrogated the TNBC cancer epithelial cells in the scRNAseq breast cancer atlas for various genes associated with luminal and basal differentiation status in breast epithelium, as well as Hallmark and MSigDB pathways related to breast differentiation. The following table summarizes our results: These results reveal a compelling trend toward loss of luminal and acquisition of basal phenotypes with age in the TNBC cancer epithelium. These data are consistent with studies that evaluated luminal/basal phenotypes in normal breast epithelium with age, particularly (6,7), showing loss of epithelial lineage fidelity and ~~with~~ [upregulation of basal] markers in normal luminal epithelial cells with age, which presage cancer susceptibility, [and with (8)] ~~upregulation of basal~~ establishing AXL as a regulator of epithelial cell plasticity in the normal mammary gland as well as driving stemness in basal breast cancer.

Although, the METABRIC analyses show enrichment of KRT6B, FOXC1, SOX10, and ELF5 in younger and enrichment of FOXA1 in older TNBC patients, none of these genes was significantly differentially expressed in the TCGA dataset. We believe this inconsistency between datasets could reflect the limitations of bulk transcriptional profiling/sequencing, as it is not clear which cells differentially express these factors in the METABRIC and TCGA datasets. For example, examination of the scRNAseq breast cancer atlas data indicates that the normal myoepithelium contributes the greatest expression of KRT14 and KRT6B signals in the young cohorts.

Reviewer 1 Response 1: In response to this, the authors evaluated luminal and basal differentiation in TNBC from the scRNAseq breast cancer atlas relative to age. They present a table that shows decreased luminal differentiation with age with a loss of FOXA1. These results are now opposite of the METABRIC findings in figure 1. The lack of significance in the TCGA data may be due to small sample size (30 vs. 37). We agree with the reviewer that different cell types contribute different markers, however the bulk of the

expression in bulk tumors should be from the tumor cells themselves, expressing either luminal or basal markers. Perhaps analysis of a larger dataset like the Fudan TNBC dataset will provide further evidence for enrichment of loss of luminal markers with age (see following comment).

To explore the lineage status hypothesis further, we analyzed the subset of luminal and basal genes in the suggested Fudan TNBC dataset. The data for all three bulk transcriptomic datasets (METABRIC, TCGA, and Fudan), as well as a pseudobulk analysis of the single-cell RNAseq data are represented in the following table:

Dataset	Cancer Epithelium (Swarbrick)		Pseudobulk all cells (Swarbrick)		METABRIC n=50 young n=63 older		TCGA n=30 young n=37 older		FUDAN TNBC n=105 young n=58 older	
	Stage		Stage		Stage		Stage		Stage	
Treatment	I-II		I-II		I-III		I-III		unspecified, nonmetastatic	
	Treatment Naive & Treated		Treatment Naive & Treated		Treatment Naive & Treated		Treatment Naive & Treated		Treatment Naive & Treated	
Tissue acquisition	Fresh/frozen resected tissue, in silico epithelial enrichment		Fresh/frozen resected tissue, no enrichment		Histopathological review, targeting high tumor cellularity		Histopathological review, targeting >60% tumor cellularity		Microdissection, targeting >50% tumor; <30% stromal tissue	
	Gene/ pathway	Avg log2FC	Adj.p	Avg log2FC	Adj.p	log2FC	Adj.p	log2FC	Adj.p	log2FC
AR	-6.6	~0	-5.43	~0	NA	NA	.75	NS	1.94	7.1E-6
FOXA1	-5.27	~0	-5.32	~0	1.19	.02	.03	NS	2.08	9.9E-6
KRT19	-1.76	~0	-2.31	~0	.11	NS	-.19	NS	.67	NS
RBM47	-1.33	1.1E-40	.26	8.7E-20	NA	NA	.19	NS	.37	.027
PROM1	-.54	NS	-1.13	2.8E-21	-.81	NS	-.85	NS	-1.41	.004
COBL	.55	5.7E-12	-.17	2.9E-4	-.26	NS	-.67	NS	.016	NS
KRT14	3.91	~0	.44	1.2E-97	-.85	NS	-1.9	NS	-.70	NS
KRT6B	3.14	~0	1.65	3.4E-179	-1.37	.02	-.29	NS	-1.07	.035
AXL	3.70	4.4E-95	-.58	1.9E-29	NA	NA	.53	NS	.053	NS
FOXC1	1.37	7.1E-99	-.28	1.3E-24	-1.23	.01	-.47	NS	-1.41	.0002
SOX10	.62	3.2E-16	-.45	NS	-1.09	.02	-1.30	NS	-1.68	.0005
ELF5	.098	4.0E-47	-1.21	NS	-1.15	.02	-.72	NS	-1.57	.0009

Orange = Luminal genes; Teal = Basal genes

Positive Avg log2FC values=enriched with age; negative values=decreased with age

While our single-cell RNA seq analysis of cancer epithelium supports the idea of an age-associated switch from luminal to basal phenotypes – consistent with prior reports in normal breast epithelium that presages cancer – this is not supported by the bulk transcriptomic datasets, including the Fudan cohort. Taken together, the findings do not support a strong conclusion about cancer cell lineage differentiation. This may reflect the limited resolution of bulk transcriptomics, where cell-type ambiguity and reliance on just a few genes make it difficult to accurately infer lineage states of heterogeneous cancer epithelium. It is also possible that disease stage and methods by which tissues are selected for analysis (see Table) impact composition of the cancer epithelium, which would be difficult to decipher without precise epithelial cell annotation provided by single-cell analysis. The results reinforce our view that

analyses using datasets with cell type resolution, that include sufficient tumor and normal epithelial cells, are the most effective methods to draw meaningful conclusions about lineage specification with age and in cancer.

We believe it is important to note that the major findings reported in our manuscript are consistent across three transcriptomic datasets: METABRIC, TCGA, and the single-cell atlas. The findings we report were identified through separate, unbiased analyses that we then cross-referenced across datasets and further validated with external, experimental evidence. Because our main conclusions are strongly supported across multiple cohorts, and the question of lineage progression with age is inconsistently supported across the datasets, we believe it would be premature to address this hypothesis in the manuscript. The possibility of exploring this line of inquiry through experimental and computational methods would be an exciting avenue for future investigation. We therefore opted to highlight the strength of this work as a hypothesis-generating Analysis manuscript and include reference to the Fudan dataset in the following revised Discussion paragraph on lines 646-659:

“A key strength of our study is the analysis of multiple independent clinical cohorts, spanning both transcriptomic and proteomic modalities. This approach enabled us to identify consistent, well-supported findings on how age at diagnosis shapes the breast TME, providing a robust resource for hypothesis generation and further investigation of age-associated changes in both cancer epithelium and stroma. For example, age-related changes in breast biology may influence the distribution of breast cancer molecular subtypes⁷⁴⁻⁷⁶. In fact, studies of aging normal breast tissue have revealed key hallmarks — such as immune functional decline and loss of lineage fidelity — that contribute to cancer development⁷⁷⁻⁷⁹. Moreover, given that menopause is a defining feature of aging, it is reasonable to hypothesize that some of our observations, particularly in ER+ breast cancers, are driven by menopausal status. Future analyses of normal breast tissue datasets^{80,81} using similar approaches as those we present here will further inform age-related cancer initiation. As single-cell atlases with clinical outcome data become available, new opportunities to study how age shapes disease progression and therapeutic response will arise, potentially explaining why younger and older patients experience worse outcomes than their middle-aged counterparts^{3,4}.”

We hope this clarification is helpful and explains why we focus on findings with support across single-cell and bulk datasets.

Reviewer 1 comment 2: The conclusions are valid however the robustness of the analysis is limited to few scRNA samples, particularly when stratifying in you young and old for the cell-cell interactions. Therobustness of the age associated transcript changes identified in METABRIC could be increasedwith validation in the TCGA (for TNBC and ER+) and the Fudan TNBC cohort (PMID: 30853353.)

Previous response 2: As suggested, we also evaluated the feasibility of incorporating the Fudan TNBC cohort into our analysis. However, the available expression data from this cohort is only provided as raw FASTQ files in the Sequence Read Archive, rather than as processed, normalized count matrices. Integrating raw FASTQ data would require reprocessing with potential batch effects and methodological discrepancies compared to the METABRIC and

TCGA datasets. Given these limitations, and the high concordance we observed between larger, multi-institutional METABRIC and TCGA, we prioritized these two robust, complementary datasets for ensuring broader relevance of our initial findings.

Reviewer 1 Response 2: We agree with the authors that analyzing fastq would be too cumbersome for the Fudan dataset. However, the authors have provided processed data on NODE. Below is the data availability statement from the manuscript. The inclusion of the Fudan cohort may provide more support for the confounding data with the shift in the luminal subtype with age.

The accession number for all the data reported in this paper is NODE: OEP000155. All data can be viewed in The National Omics Data Encyclopedia (NODE) (<http://www.biosino.org/node>) by pasting the accession (OEP000155) into the text search box or through the URL: <http://www.biosino.org/node/project/detail/OEP000155>

This assistance is greatly appreciated and enabled us to easily perform the requested analyses. The conclusions from these analyses are mentioned in the comment above.

Reviewer #2 (Remarks to the Author):

The authors have thoroughly addressed my previous concerns. They have expanded the dataset, clarified methodological details, and provided additional analyses and explanations that strengthen the conclusions. I appreciate the thoughtful revisions, which improve the clarity, rigor, and overall impact of the study.

Thank you for your time and thoughtful consideration of the revised manuscript; we are pleased to know we addressed your previous concerns.

Reviewer #2 (Remarks on code availability):

The code is well prepared, clearly documented, and facilitates reproducibility.

Thank you.

Reviewer #3 (Remarks to the Author):

The authors have made substantial and commendable revisions in response to the initial round of reviews. Notably, they have increased the sample size and incorporated two independent clinical cohorts to validate key findings derived from the Aspen algorithm. Furthermore, they have strengthened their conclusions by employing multiplex fluorescence imaging on tumor tissue from an additional independent cohort, which includes both older and younger women with triple-negative breast cancer (TNBC) or ER-positive disease. These additions significantly enhance the robustness and relevance of the study. Overall, I am impressed by the depth and rigor of the revisions.

However, I would like to draw attention to the discussion section, specifically lines 646–660. The authors still acknowledge the limitations of their sample size, though they do not adequately emphasize the fact that they have addressed this concern through

validation in orthogonal cohorts. This is a critical strength of the revised manuscript and should be highlighted more explicitly. Doing so would not only reinforce the credibility of their findings but also provide a valuable roadmap for other researchers facing similar challenges in validating gene-level predictions in limited-sample contexts.

Response: Thank you for acknowledging the strength of the revised manuscript and the helpful suggestion to emphasize its contribution to the field. We revised the Discussion in lines 646-649 as follows:

“A key strength of our study is the analysis of multiple independent clinical cohorts, spanning both transcriptomic and proteomic modalities. This approach enabled us to identify consistent, well-supported findings on how age at diagnosis shapes the breast TME, providing a robust resource for hypothesis generation and further investigation of age-associated changes in both cancer epithelium and stroma.”